# GenAI vs. Human Creators: Procurement Mechanism Design in Two-/Three-Layer Markets

**Rui Ai**                                                              *ruiai@mit.edu*
*Massachusetts Institute of Technology*
**David Simchi-Levi**                                                   *dslevi@mit.edu*
*Massachusetts Institute of Technology*
**Haifeng Xu**                                                   *haifengxu@uchicago.edu*
*The University of Chicago*

**Reviewed on OpenReview:** *https://openreview.net/forum?id=Eukf4TBHS7*

## Abstract

With the rapid advancement of generative AI (GenAI), mechanism design adapted to its unique characteristics poses new theoretical and practical challenges. Unlike traditional goods, content from one domain can enhance the training and performance of GenAI models in other domains. For example, OpenAI's video generation model Sora (Liu et al., 2024b) relies heavily on image data to improve video generation quality. In this work, we study *nonlinear procurement mechanism design* under *data transferability*, where online platforms employ both human creators and GenAI to satisfy cross-domain content demand. We propose optimal mechanisms that maximize either platform revenue or social welfare and identify the specific properties of GenAI that make such high-dimensional design problems tractable. Our analysis further reveals which domains face stronger competitive pressure and which tend to experience overproduction. Moreover, the growing role of data intermediaries, including labeling companies such as Scale AI and creator organizations such as The Wall Street Journal, introduces a third layer into the traditional platform–creator structure. We show that this three-layer market can result in a lose-lose outcome, reducing both platform revenue and social welfare, as large pre-signed contracts distort creators' incentives and lead to inefficiencies in the data market. These findings suggest a need for government regulation of the GenAI data ecosystem, and our theoretical insights are further supported by numerical simulations.

## 1 Introduction

Large models have entered the era of multimodality (Yin et al., 2023). Modern commercial systems such as GPT-4 (Achiam et al., 2023), Gemini (Team et al., 2023), and Claude (Anthropic, 2024) are trained on diverse types of data, including text, images, audio, and video. These modalities are not isolated; transferability among them is well established both theoretically and empirically through transfer learning (Howard & Ruder, 2018). Just as film directors may draw inspiration from Shakespeare's works, image datasets can enhance the performance of video generation models. This interdependence links the valuation of data across domains into a unified economic problem and naturally raises the question:

*How do we price content across different domains?*

The answer depends on the degree of transferability, a feature absent from traditional goods. Bricks, for instance, can only build houses but not chips, while data can improve the performance of models across tasks. Data with high transferability should therefore command a higher price, whereas specialized data that benefits only a single task should be valued lower. In this paper, we incorporate parameters capturing data transferability into a formal production model to analyze how this property shapes market outcomes.

The transferability of data also raises important ethical concerns. Recent debates have centered on whether GenAI will cause certain professions to vanish, alongside growing discussions on the implications of artificial general intelligence (George et al., 2023). We want to ask:

*From an economic perspective, will any domains disappear?*

In Section 2.2, we show that although GenAI may reduce employment in some domains, it cannot completely replace them. This result follows from the principles of diminishing marginal returns and increasing marginal costs. Moreover, in domains with high data transferability, we observe a tendency toward overproduction, which distorts the allocation of creative effort. Such imbalances highlight the need for regulatory oversight and further research on how GenAI reshapes data generation markets.

The growing demand for data to train large models has driven many AI companies to purchase high-quality datasets. For example, OpenAI reportedly spent about $250 million acquiring data from The Wall Street Journal (WSJ, 2024). Industry forecasts further predict that the global generative AI market for content creation will expand from USD 15.2 billion in 2024 to USD 175.3 billion by 2033 (Market.us, 2024). This surge in data demand can be traced back to the creation of ImageNet (Deng et al., 2009), which catalyzed the modern data economy. The emergence of specialized data annotation companies such as Scale AI, Lionbridge, Aurora AI, and Amazon Mechanical Turk has since transformed the landscape, pushing the market from a two-layer structure toward a three-layer structure that includes professional data brokers. This evolution raises a key question:

*How does the three-layer market impact revenue and social welfare?*

In Section 3, we examine two types of platforms—those that maximize revenue and those that maximize social welfare—and show that in the three-layer market, both platform revenue and social welfare decline relative to the two-layer setting, resulting in a clear lose-lose outcome. The presence of data brokers distorts creators' incentives and reduces overall efficiency, highlighting the need for regulatory oversight to address inefficiencies in the data market.

## 1.1 Our Contributions

We summarize our contributions in three main aspects and elaborate on each below.

**Mechanism design with competition between humans and generative AI.** Recent studies have begun examining market equilibria that arise from competition between AI content generators and human creators (Yao et al., 2024). Our work is the first to approach this problem from the buyer's perspective. We study online sharing platforms that rely on both human creators and GenAI for content production, situating the analysis within the broader framework of procurement mechanism design. Going beyond the classical single-dimensional setting (Myerson, 1981), we derive optimal mechanisms for five of six multi-dimensional environments and establish tight upper and lower bounds for revenue and social welfare in the remaining case. Our results show that although GenAI reduces the overall demand for human-created content, human creators remain indispensable from an economic standpoint. Their outputs not only satisfy subscriber preferences but also provide essential data for GenAI training. Furthermore, in domains with high data transferability, we find that overproduction is more likely to occur, reflecting the complex interactions between human and algorithmic production incentives.

**Distinguishing between two/three-layer markets regarding data brokers.** Finding an efficient market structure has been a long-standing goal in digital economics (Liang et al., 2018; Agarwal et al., 2019). For online platforms, a central question is whether to buy content directly from human creators as crowdsourcing, or through data companies. Our analysis shows that the latter, a three-layer market where the platform purchases data from a broker who sources it from humans, leads to a lose-lose outcome: both platform revenue and social welfare decline. The inefficiency stems from asymmetric information. Large, transparent contracts reveal the platform's and broker's valuations to creators, making price discrimination impossible and weakening incentives for efficient production. As a result, both welfare and revenue fall, underscoring the need for policy intervention.

**Numerical experiments validate the conclusions about market dynamics.** Finally, we validate our theoretical results using simulated data. In the two-layer market, the experiments reveal content overproduction in specific domains. We also compare platform revenue and social welfare across the two- and three-layer settings, illustrating the "lose-lose" outcome in the latter. These numerical experiments offer valuable inspiration and insights for future real-world mechanism design and implementation using real data.

## 1.2 Related Works

Our work is closely related to the literature on *procurement mechanism design* (Myerson, 1981; McAfee & McMillan, 1987; Laffont & Tirole, 1993), *data pricing* (Jia et al., 2019; Ghorbani & Zou, 2019; Schoch et al., 2022; Ai et al., 2024), *human vs. GenAI competition* (Esmaeili et al., 2024; Yao et al., 2024; Fish et al., 2024; Duetting et al., 2024) and *multi-layer markets* (Fallah et al., 2024; Xu et al., 2020). We generalize physical goods procurement to transferable digital content and analyze the ensuing human–GenAI competition across multi-layer markets. A thorough comparison and additional related work are provided in Appendix A.1.

# 2 Content Procurement in (Two-Layer) Platform-Creator Markets

We begin with the two-layer market, in which online platforms procure content directly from human creators. For example, platforms such as YouTube invite creators to upload content in exchange for monetary rewards, providing a representative case of this market structure.

## 2.1 A Model of Content Procurement in the GenAI Era

An online platform features $K$ domains (e.g., sports, pop music, knowledge sharing), denoted by $[K] = 1, 2, \ldots, K$. To serve its user base, the platform requires $d_k \in \mathbb{R}$ units of content in each domain $k \in [K]$, produced by human creators and/or GenAIs. We assume that each human creator specializes in a single domain[1]. For instance, @dream has attracted over 30 million YouTube subscribers through Minecraft challenge videos. For analytical convenience, each domain $k$ is represented by a single creator who incurs a cost $c_k x_k^{\rho_k}$ to produce $x_k$ units of content, following standard models of the creator economy (Hu et al., 2023; Jagadeesan et al., 2024; Yao et al., 2024). We interpret $x_k$ as a *calibrated quantity*, meaning that production is weighted by quality metrics such as click-through rate or content accuracy (Radlinski et al., 2008). The parameter $c_k$ captures the creator's *production efficiency*, which is private information, while $\rho_k$ reflects the population-level growth speed of the production cost and is assumed to be publicly known, as it can be inferred from market data. This representative-creator assumption is without loss of generality: when multiple creators exist in the same domain with heterogeneous cost parameters $c_k$, they can be aggregated into a single "meta" creator with an effective cost coefficient $c_k$ (see Appendix C.4 for details). Consistent with prior work, we assume $\rho_k \geq 1$ to reflect increasing marginal costs (Marshall, 2009). This captures the empirical observation that creative ideas become progressively harder to generate over given time, resulting in a convex production cost function.

To meet its demand requirements $(d_1, \ldots, d_K)$, the platform (e.g., a video-sharing site such as YouTube) asks human creators to report their private cost parameters $c = (c_1, \ldots, c_K)$. To account for possible misreporting, let $\widehat{c} = (\widehat{c}_1, \ldots, \widehat{c}_K)$ denote the reported costs. The mechanism then proceeds according to the standard timeline in mechanism design (Hart & Tirole, 1988; Salant, 1989):

1. The platform commits to a mechanism $x(\cdot)$ and $p(\cdot)$;
2. The creators' costs $\widehat{c}$ are then elicited;
3. The platform implements "allocation outcomes" $x(\widehat{c}) = (x_1(\widehat{c}), ..., x_K(\widehat{c}))$ by procuring $x_k(\widehat{c})$ amount of domain-$k$ content from creator $k$ for each $k$;
4. The payments $p(\widehat{c}) = (p_1(\widehat{c}), ..., p_K(\widehat{c}))$ is then executed by paying $p_k(\widehat{c})$ to creator $k$.

While procurement auctions have been extensively studied in prior work (Laffont & Tirole, 1993), our setting differs by incorporating the platform's ability to use GenAI to augment content creation at negligible cost

---

[1] Although some creators may produce content across domains, most major creators focus on one area in practice; see Figure 1 for examples.

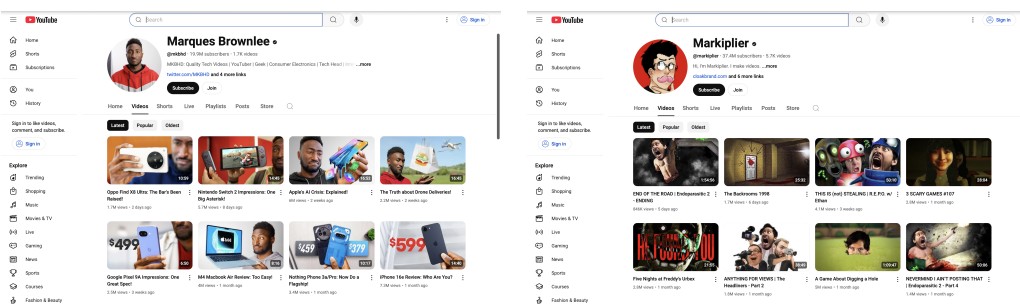

Figure 1: @mkbhd focuses on tech reviews, and @markiplier is dedicated to gaming content, especially horror-themed Let's Plays.

(We discuss the non-negligible case in Appendix C.1). Specifically, we assume that the platform can employ GenAI, together with human-created content, to partially meet demand with minimal additional expense relative to human creation. Motivated by the scaling laws of GenAI's creative capacity (Kaplan et al., 2020; Yao et al., 2024), we model this process as follows: $x_i$ units of content in domain $i$ enable GenAI to generate $\mu_{ik} x_i^{\gamma_{ik}}$ units of essentially new content in domain $k$, where $\mu_{ik}$ captures the transferability from domain $i$ to domain $k$, and $\gamma_{ik}$ represents the capability of employed GenAI tool. Yao et al. (2024) adopts a similar assumption to analyze the symbiosis or conflict between GenAI and human creators. This formulation highlights three key distinctions between GenAI, human creators, and traditional machines. First, GenAI exhibits transferability, allowing knowledge from one domain to benefit others with varying efficiency. Second, its marginal cost of production is negligible compared with human effort. Third, GenAI's productivity depends on human creation, unlike traditional machines. The power-law relationship between model capability and training data is well documented in the GenAI literature (Kaplan et al., 2020), and we assume that these capability parameters are known, as they can be empirically estimated (Alabdulmohsin et al., 2022; Goyal et al., 2024; Lin et al., 2024). Finally, as is standard, we restrict $\gamma_{ik} \in [0,1]$ to reflect the diminishing efficiency of data. Accordingly, given reported costs $\widehat{c}$, the platform's design must satisfy the following demand constraints:

$$\text{Demand: } x_k(\widehat{c}) + \sum_{i=1}^{K} \mu_{ik} x_i(\widehat{c})^{\gamma_{ik}} \geq d_k \text{ for any } k \in [K].$$

The revelation principle (Roughgarden, 2010) implies that, without loss of generality, we can restrict attention to incentive-compatible (IC) mechanisms in which truthful reporting constitutes a Nash equilibrium (Myerson, 1986). In Appendix A.2, we provide illustrations about the economic background for these and other relevant terms. Accordingly, the incentive-compatibility condition is given by

$$\text{IC: } \mathbb{E}_{-k}[-c_k x_k^{\rho_k}(c) + p_k(c)] \geq \mathbb{E}_{-k}[-c_k x_k^{\rho_k}(\widehat{c}_k, c_{-k}) + p_k(\widehat{c}_k, c_{-k})] \text{ for any } k \in [K],$$

where $c_{-k}$ is $c$ except the $k$-th entry. The expectation is taken over domains $i \neq k$. The nonlinearity of the cost function hinders the application of existing methods (Myerson, 1981). We first focus on the Nash equilibrium (Nash Jr, 1950), assuming no collusion, so it suffices to consider one-shot deviations. We will later extend the analysis to more complex settings involving creator union and data brokers. An interesting future direction is to extend it to correlated equilibrium and coarse correlated equilibrium (Aumann, 1987) with limited signals.

Moreover, since human creators can always reject any allocation–payment pair offered by the platform and quit the market, the mechanism must also satisfy the individual rationality (IR) condition:

$$\text{IR: } \mathbb{E}_{-k}[-c_k x_k^{\rho_k}(c) + p_k(c)] \geq 0 \text{ for any } k \in [K].$$

We consider the following class of allocation rules, which are natural when items are substitutable, as in content production. In a valid mechanism, a creator's production $x_k$ decreases with her own cost and increases weakly as the costs of other creators rise.

**Definition 2.1** (Valid Mechanisms). We say a mechanism is valid if its allocation rule satisfies

- $x_k(c_k, c_{-k}) \geq x_k(\widetilde{c}_k, c_{-k})$ for any $\widetilde{c}_k \geq c_k$ and
- $x_k(c_k, c_{-k}) \leq x_k(c_k, \widetilde{c}_{-k})$ if $\widetilde{c}_i \geq c_i$ for any $i \neq k$.

In practice, platforms typically pursue one of two objectives. Some aim to maximize revenue, or equivalently, minimize total cost. Others, such as large technology firms like Google (Google, 2024) and Meta (Meta, 2024), emphasize social welfare, defined as the aggregate utility of all market participants, or equivalently, the minimization of creators' total costs, since monetary transfers do not affect overall welfare. These considerations motivate the two optimization objectives:

- Type 1: $\min_{x,p} \mathbb{E}_c[\sum_{k=1}^{K} p_k(c)]$;
- Type 2: $\min_{x,p} \mathbb{E}_c[\sum_{k=1}^{K} c_k x_k^{\rho_k}(c)]$.

For the second objective, when multiple mechanisms yield the same level of social welfare, we select the one with the minimum total payment, breaking ties in favor of the platform.

For simplicity, we impose the following assumption on the distribution of cost parameter $c$.

**Assumption 2.1** (Independent cost). We assume that the cost components $c_1, ..., c_K$ are independent with p.d.f. $f_1(c_1), ..., f_K(c_K)$ and corresponding c.d.f. as $F_1(c_1), ..., F_K(c_K)$, respectively. Additionally, we define $f(c) = \prod_{k=1}^{K} f_k(c_k)$, $F(c) = \prod_{k=1}^{K} F_k(c_k)$, $f_{-i}(c_{-i}) = \prod_{k \neq i} f_k(c_k)$ and $F_{-i}(c_{-i}) = \prod_{k \neq i} F_k(c_k)$.

To avoid lengthy discussions of corner cases, we impose standard regularity assumptions on the above distributions, as commonly adopted in both the economics and machine learning literature (Myerson, 1981; Wang et al., 2019; Ai et al., 2022).

**Assumption 2.2** (Bounded cost and density). We assume that for any $k \in [K]$, $c_k$ is bounded, say $c_k \in [a_k, b_k]$, and $f_k(c_k)$ is lower bounded from 0.

**Assumption 2.3** (Monotone reverse hazard rate). We assume that for all $k \in [K]$, it holds that $F_k(\cdot)$ is log-concave, in other words, the ratio $\frac{f_k(\cdot)}{F_k(\cdot)}$ is monotone decreasing.

Assumption 2.2 can always be ensured through distribution truncation, while Assumption 2.3 is a standard and widely popular assumption in economics (Kleiber, 2003; Bagnoli & Bergstrom, 2006; Wang et al., 2024). These assumptions are mild and hold for many commonly used distributions, such as the truncated Gaussian and the uniform distribution (Golrezaei et al., 2019).

## 2.2 The Optimal Mechanism $\mathcal{M}_1$ for Revenue-Maximizing Procurement

We first consider the case in which the platform aims to maximize its own revenue. Accordingly, the platform faces the following revenue-maximization (or cost-minimization) problem:

$$
\begin{aligned}
Rev_1 = \max_{x,p} \quad & \mathbb{E}_c[-\sum_{k=1}^{K} p_k(c)] \\
\text{s.t.} \quad & x_k(c) + \sum_{i=1}^{K} \mu_{ik} x_i(c)^{\gamma_{ik}} \geq d_k \text{ and } x_k(c) \geq 0 \text{ for any } k \in [K] \\
& \mathbb{E}_{-k}[-c_k x_k^{\rho_k}(c) + p_k(c)] \geq \mathbb{E}_{-k}[-c_k x_k^{\rho_k}(\widehat{c}_k, c_{-k}) + p_k(\widehat{c}_k, c_{-k})] \\
& \mathbb{E}_{-k}[-c_k x_k^{\rho_k}(c) + p_k(c)] \geq 0,
\end{aligned}
\tag{1}
$$

where the expectation is taken over $c \sim F(c)$. In addition, the corresponding social welfare is $SW_1 = \mathbb{E}_c[-\sum_{k=1}^{K} c_k x_k^{\rho_k}(c)]$. We now present the optimal mechanism $\mathcal{M}_1$ under information asymmetry between human creators and the platform in Algorithm 1.

**Theorem 2.1.** *Mechanism $\mathcal{M}_1$ is a valid mechanism satisfying IC and IR conditions[2] and achieves the highest expected revenue/lowest cost for the platform.*

While Theorem 2.1 is formulated in a single-parameter mechanism design setting, the main challenge in its proof arises from the *nonlinearity* of the designer's allocation constraint, namely $y_k^{1/\rho_k} + \sum_{i=1}^{K} \mu_{ik} y_i^{\gamma_{ik}/\rho_i} \geq d_k$ in Algorithm 1. This constraint, induced by the nonlinear cost function and data transferability, prevents the direct application of the standard analysis by Myerson (1981). We address this difficulty in two steps. First,

---

[2]Since Algorithm 1 is a valid mechanism, it actually satisfies ex-post IC and IR conditions.

---

**Algorithm 1** Mechanism $\mathcal{M}_1$ for revenue-maximizing platforms without union in the two-layer market.

---

**Input:** Report $\widehat{c}$.

Calculate virtual cost: $vc_k(\widehat{c}) = \widehat{c}_k + \frac{F_k(\widehat{c}_k)}{f_k(\widehat{c}_k)}$ for any $k \in [K]$.

Call some oracle to solve the auxiliary optimization problem

$$y = \underset{y}{\operatorname{argmin}} \sum_{k=1}^{K} vc_k(\widehat{c}) y_k$$

$$\text{s.t. } y_k^{1/\rho_k} + \sum_{i=1}^{K} \mu_{ik} y_i^{\gamma_{ik}/\rho_i} \geq d_k \text{ and } y_k \geq 0 \text{ for any } k \in [K].$$

Calculate the allocations: $x_k(\widehat{c}) = y_k^{1/\rho_k}$ for any $k \in [K]$.

Calculate the payments: $p_k(\widehat{c}) = \widehat{c}_k y_k + \int_{c_k}^{b_k} y_k(\widehat{c}_1, ..., \widehat{c}_{k-1}, t_k, \widehat{c}_{k+1}, ..., \widehat{c}_K) dt_k$ for any $k \in [K]$.

**Output:** Allocation-payment pair $(x(\widehat{c}), p(\widehat{c})) = ((x_1(\widehat{c}), ..., x_K(\widehat{c})), (p_1(\widehat{c}), ..., p_K(\widehat{c})))$.

---

by applying a nonlinear transformation, we replace the search for the optimal allocation $x(\cdot)$ with that for $y(\cdot)$, a monotonic transformation of $x(\cdot)$, which linearizes the cost term. Second, we show that the nonlinear constraints introduced by GenAI still preserve convexity. Together, these steps reformulate Problem 1 into a convex optimization problem, for which we establish the existence of an efficient solution algorithm.

**Proposition 2.1.** *Algorithm 1 can be implemented to output an $\epsilon$-optimal mechanism[3] in $\mathcal{O}(\frac{1}{\epsilon^4})$ time. Furthermore, if $\max_i \{\frac{\gamma_{ik}}{\rho_i}\} < 1$ for any $k \in [K]$, the computational complexity reduces to $\mathcal{O}(\frac{1}{\epsilon^3})$.*

Randomized payments can further reduce the computational complexity to $\mathcal{O}(1/\epsilon^2)$ and $\mathcal{O}(1/\epsilon)$, as detailed in Proposition C.1 of Appendix C.2. These results demonstrate that online platforms can efficiently maximize revenue even when incorporating GenAI into content creation. This extends the scope of procurement mechanism design and provides a theoretical foundation for pricing in the rapidly evolving GenAI economy.

Furthermore, we derive the following corollary from Theorem 2.1, which shows that original human-created content remains essential in every domain and that complete substitution does not occur. However, unlike markets without GenAI, overproduction may emerge as a new phenomenon.

**Corollary 2.2.** *Assuming data has transferability, i.e., all $\mu_{ik}$ and $\gamma_{ik}$ are not zero, then no domain will disappear, i.e., all $x_k(c)$ will be positive, no matter the value of $c$. However, some demand constraints will be non-binding, i.e., some $x_k(c) + \sum_{i=1}^{K} \mu_{ik} x_i(c)^{\gamma_{ik}} > d_k$. It shows that with the development of GenAI, some domains will be overproduced to augment knowledge transfer.*

## 2.3 The Optimal Mechanism $\mathcal{M}_2$ for Welfare-Maximizing Procurement

To maximize social welfare, an intuitive way is to allocate $x$ corresponding to $\operatorname{argmin}_x \sum_k c_k x_k^\rho(c)$ for each $c$. This yields the following optimization formulation and the corresponding revenue for such kind of platforms is $Rev_2 = \mathbb{E}_c[-\sum_{k=1}^{K} p_k(c)]$.

$$
\begin{aligned}
SW_2 = \max_{x,p} \quad & \mathbb{E}_c[-\sum_{k=1}^{K} c_k x_k^{\rho_k}(c)] \\
\text{s.t.} \quad & x_k(c) + \sum_{i=1}^{K} \mu_{ik} x_i(c)^{\gamma_{ik}} \geq d_k \text{ and } x_k(c) \geq 0 \text{ for any } k \in [K] \\
& \mathbb{E}_{-k}[-c_k x_k^{\rho_k}(c) + p_k(c)] \geq \mathbb{E}_{-k}[-c_k x_k^{\rho_k}(\widehat{c}_k, c_{-k}) + p_k(\widehat{c}_k, c_{-k})] \\
& \mathbb{E}_{-k}[-c_k x_k^{\rho_k}(c) + p_k(c)] \geq 0.
\end{aligned}
\tag{2}
$$

However, two key challenges arise in deriving the optimal design: (a) incentivizing each creator to report their cost truthfully, and (b) identifying a payment rule $p(\cdot)$ that satisfies both IR and validity constraints. We resolve both challenges in the affirmative and derive the optimal mechanism $\mathcal{M}_2$ for this setting by substituting $vc(\widehat{c})$ with $\widehat{c}$. Details are provided in Algorithm 4 of Appendix B.

---

[3]It means compared to the optimal mechanism, the extra loss is at most $\epsilon$.

We conclude this section with our main result, which establishes the optimality of Algorithm 4, i.e., mechanism $\mathcal{M}_2$. The availability of the optimal mechanism also explains the rationale for the presence of companies in the market that pursue social welfare maximization.

**Theorem 2.2.** *Mechanism $\mathcal{M}_2$ is a valid mechanism satisfying IC and IR conditions and achieves the highest social welfare with polynomial time complexity[4].*

### 2.4 Creator Union and Optimal Procurement Mechanism Design

Recently, OpenAI and The Wall Street Journal reached a data agreement valued at up to $250 million (WSJ, 2024). At the same time, data-selling companies such as Scale AI have experienced rapidly rising valuations, driven by the growing demand for high-quality data to fine-tune large language models (The Scale Team, 2023). Economically, these intermediaries can be viewed as aggregating data from human creators and repackaging it for sale, similar to a creator union. We defer the analysis of the case in which such intermediaries secure contracts before sourcing data from creators to Section 3. This raises a key question: when a creator union appears, represented by a single agent producing content across all $K$ domains, how will the revenue of data-obsessing platforms and social welfare be affected?

We retain the setting from Section 2.1, except that the representative creator now has a cost vector $c = (c_1, \ldots, c_K)$. We also let $a = (a_1, \ldots, a_K)$ and $b = (b_1, \ldots, b_K)$. Unlike before, one-shot deviations are insufficient to capture strategic behavior, as the creator may misreport multiple entries simultaneously. This possibility of collusion complicates the mechanism design problem by introducing correlations, analogous to the distinction between Nash and correlated equilibria (Gilboa & Zemel, 1989).

#### 2.4.1 Revenue-Maximizing Procurement Mechanism $\mathcal{M}_3$ with a Creator Union

Pessimistically, general high-dimensional mechanism design problems remain open and agnostic (Briest et al., 2010; Hart et al., 2013; Daskalakis, 2015; Hart & Nisan, 2017). Therefore, we formulate the corresponding optimization problem and derive bounds on platform revenue and social welfare, leaving explicit analytical solutions for future research. The revenue-maximizing platform now faces the following optimization problem (Mechanism $\mathcal{M}_3$), where the payment rule $p(\cdot)$ is a scalar function and the corresponding social welfare is $SW_3 = \mathbb{E}_c[-\sum_{k=1}^{K} c_k x_k^{\rho_k}(c)]$:

$$
\begin{aligned}
Rev_3 = \max_{x,p} \quad & \mathbb{E}_c[-p(c)] \\
\text{s.t.} \quad & x_k(c) + \sum_{i=1}^{K} \mu_{ik} x_i(c)^{\gamma_{ik}} \geq d_k \text{ and } x_k(c) \geq 0 \text{ for any } k \in [K] \\
& p(c) - \sum_{k=1}^{K} c_k x_k^{\rho_k}(c) \geq p(\widehat{c}) - \sum_{k=1}^{K} c_k x_k^{\rho_k}(\widehat{c}) \\
& p(c) - \sum_{k=1}^{K} c_k x_k^{\rho_k}(c) \geq 0.
\end{aligned}
\tag{3}
$$

The main challenge in this problem is that the union may misreport multiple components of the cost vector $c$ simultaneously. The resulting correlations make the optimization problem effectively contain infinitely many constraints and require the use of path integrals in defining the payment rule. Finding the optimal solution remains a difficult open problem in economics. To be specific, recall in the one-dimensional case, the optimal payment in Algorithm 1 has the form $p_k = c_k y_k + \int_c^b y_k(c_1, \ldots, c_{k-1}, t_k, c_{k+1}, \ldots, c_K) dt_k$ with truthful reports. However, in the high-dimensional case, the corresponding payment would become $p = c \cdot y + \int_c^b y(t) \cdot t$[5]. When $c$ is a high-dimensional vector, it raises an extra question that the integration results might be different for different paths from $c$ to $b$, which contradicts the fixed contract. Hence, simply applying methods in the one-dimensional case won't yield a valid mechanism, and it hinders the optimal high-dimensional mechanism design in relevant scenarios. In contrast, as shown in the next section, a platform that maximizes social welfare can efficiently obtain the optimal solution, highlighting a notable advantage.

---

[4]In application, it is also known as Fully Polynomial-Time Approximation Scheme (FPTAS), which means that we can find an $\epsilon$-optimal solution within polynomial time with respect to $1/\epsilon$.

[5]The symbol $\cdot$ represents the dot product.

### 2.4.2 Welfare-Maximizing Procurement Mechanism $\mathcal{M}_4$ under a Creator Union

The main obstacle to obtaining explicit solutions in high-dimensional mechanism design lies in the IC condition. To ensure truthful reporting by the creator union, the payment rule must involve a high-dimensional integral, and it is generally difficult to construct one that is independent of the integration path.

In this section, we turn to studying the properties of social welfare maximizers who aim to minimize total social cost $\mathbb{E}_c[\sum_k c_k x_k^{\rho_k}(c)]$. Similarly, we use $Rev_4 = \mathbb{E}_c[-p(c)]$ to denote corresponding platform revenue.

$$
\begin{aligned}
SW_4 = \max_{x,p} \quad & \mathbb{E}_c[-\sum_k c_k x_k^{\rho_k}(c)] \\
\text{s.t.} \quad & x_k(c) + \sum_{i=1}^{K} \mu_{ik} x_i(c)^{\gamma_{ik}} \geq d_k \text{ and } x_k(c) \geq 0 \text{ for any } k \in [K] \\
& p(c) - \sum_{k=1}^{K} c_k x_k^{\rho_k}(c) \geq p(\widehat{c}) - \sum_{k=1}^{K} c_k x_k^{\rho_k}(\widehat{c}) \\
& p(c) - \sum_{k=1}^{K} c_k x_k^{\rho_k}(c) \geq 0.
\end{aligned}
$$

Nonetheless, the analysis shows that the optimal mechanism can be derived in an explicit form and computed in polynomial time. This may explain why some companies choose to maximize social welfare, as the corresponding mechanism is easier for customers to understand and involves only a small loss in revenue. We now present the optimal mechanism $\mathcal{M}_4$ under this setting in Algorithm 2.

---

**Algorithm 2** Mechanism $\mathcal{M}_4$ for Type 2 platforms with union in the two-layer market.

---

**Input:** Report $\widehat{c}$.
Replace virtual cost: $vc(\widehat{c})$ by $\widehat{c}$.
Call Algorithm 1 for $y$.
Calculate the allocations: $x_k(\widehat{c}) = y_k^{1/\rho_k}$ for any $k \in [K]$.
Calculate the payment: $p(\widehat{c}) = \widehat{c} \cdot y + \int_{\widehat{c}}^{b} y(t) \cdot dt$.
**Output:** Allocation-payment pair $(x(\widehat{c}), p(\widehat{c})) = ((x_1(\widehat{c}), ..., x_K(\widehat{c})), p(\widehat{c}))$.

---

Note that calculating the payment involves an integration path. A natural question is whether $p(\cdot)$ is well-defined, that is, independent of the chosen path. We answer the question in the following Lemma 2.3.

**Lemma 2.3.** *The payment $p(\cdot)$ in Algorithm 2 is path independent and so well-defined.*

We now state the main theorem establishing the optimality of our Algorithm 2.

**Theorem 2.3.** *Mechanism $\mathcal{M}_4$ is a valid mechanism satisfying IC and IR conditions and achieves the highest social welfare with polynomial time complexity.*

This theorem reveals a surprising result: despite the high-dimensional setting, the welfare-maximizing mechanism can be computed efficiently. The proof is constructive, identifying an optimal payment rule $p(\cdot)$ that is independent of the integration path. The distinctive properties of GenAI are crucial to this result. Since GenAI introduces only convex demand constraints, the allocation, whether at a vertex or along a smooth segment, remains orthogonal to cost after transformation. We construct a potential function $\Psi(c)$, detailed in Appendix C.6, and establish the relationship between its gradient and the allocation rule. This approach provides a new method for addressing high-dimensional mechanism design problems, distinct from cycle monotonicity (Lavi & Swamy, 2007), and advances understanding of this open question. Building on the above results, the next section presents a partial comparison of $Rev_{1-4}$ and $SW_{1-4}$.

### 2.5 Comparison of Revenue and Social Welfare for Mechanisms $\mathcal{M}_{1-4}$

Intuitively, a revenue-maximizing platform (Type 1) should generate lower social welfare than a welfare-maximizing platform (Type 2) under the same setting, while a Type 2 platform typically earns less revenue. Regarding the creator union, since it aggregates information from all human creators, it is expected to possess greater market power, potentially leading to higher total utility for creators. The following theorem formalizes these comparison results.

**Theorem 2.4.** *Let $Rev_i$ and $SW_i$ denote the revenue and welfare of valid mechanism $\mathcal{M}_i$. For the revenue objective, we have $Rev_1 \geq Rev_3$ and $Rev_1 \geq Rev_2 \geq Rev_4$. For the social welfare objective, we have $SW_2 = SW_4 \geq \max\{SW_1, SW_3\}$.*

The most technically interesting comparison lies between $Rev_2$ and $Rev_4$. Since the payment rule of $\mathcal{M}_4$ involves a path integral, we select a specific integration path and, using the validity of the mechanism, show that $Rev_2 \geq Rev_4$. The complete proof is provided in Appendix C.7.

As shown in Theorem 2.4, we establish the comparative rankings of revenue and social welfare among Algorithms 1, 2 and 4, and provide upper bounds for $\mathcal{M}_3$ under mild assumptions, since it lacks an explicit solution. We find that the presence of a creator union reduces platform revenue by weakening its pricing power. In contrast, for a social welfare–maximizing platform, the union does not affect the outcome. Intuitively, social welfare depends solely on the creators' total production cost, so while the union may alter the payment rule, it has no incentive to distort the allocation. As social welfare is allocation-dependent only, its optimality remains preserved.

# 3 Content Procurement in Platform-Broker-Creator Markets

In this section, we extend our analysis beyond the union case and consider a three-layer market (Fallah et al., 2024). In this setting, the platform first offers a contract $(z, t)$ to a data broker (Liu et al., 2021), who decides whether to accept it. The broker then engages with human creators and faces a mechanism design problem similar to that in Section 2.1. Unlike the union, which maximizes the total utility of creators, the broker focuses solely on profit, defined as the difference between $z$ and $\sum_{k=1}^{K} p_k$. In practice, companies such as SchoolDigger and Datarade serve as examples of data brokers (Zhang et al., 2024a). To build intuition, Figure 7 in Appendix D.1 illustrates the timeline of this mechanism design setting.

## 3.1 Revenue-Maximizing Procurement Mechanism $\mathcal{M}_5$ with a Data Broker

We first consider the scenario in which the platform seeks to maximize its revenue. To identify the optimal mechanism, we begin by formulating the broker's optimization problem given $(z, t)$. Since both the allocation and payment rules depend on $(z, t)$, we denote them by $x(\cdot; z, t)$ and $p(\cdot; z, t)$, respectively. Because the broker signs the contract with the platform in advance, the credibility constraint (Akbarpour & Li, 2020) imposes an additional condition that, for any report $\widehat{c}$,

$$x_k(\widehat{c}) \geq z_k \text{ for any } k \in [K].$$

Consequently, the broker needs to solve the following optimization problem:

$$
\begin{aligned}
\max_{x,p} \quad & \mathbb{E}_c[t - \sum_{k=1}^{K} p_k(c; z, t)] \\
\text{s.t.} \quad & \mathbb{E}_{-k}[-c_k x_k^{\rho_k}(c; z, t) + p_k(c; z, t)] \geq \mathbb{E}_{-k}[-c_k x_k^{\rho_k}(\widehat{c}_k, c_{-k}; z, t) + p_k(\widehat{c}_k, c_{-k}; z, t)] \\
& \mathbb{E}_{-k}[-c_k x_k^{\rho_k}(c; z, t) + p_k(c; z, t)] \geq 0 \\
& x_k(c; z, t) \geq z_k \text{ for any } k \in [K],
\end{aligned}
\tag{4}
$$

where $t$ is a fixed constant and $z$ is a fixed $K$-dimensional nonnegative vector. Let $x^*(\cdot; z, t)$ and $p^*(\cdot; z, t)$ denote the optimal allocation and payment rules of Problem 4. The platform's objective is then given by

$$
\begin{aligned}
Rev_5 = \max_{z,t} \quad & \mathbb{E}_c[-t] \\
\text{s.t.} \quad & z_k + \sum_{i=1}^{K} \mu_{ik} z_i^{\gamma_{ik}} \geq d_k \text{ and } z_k \geq 0 \text{ for any } k \in [K] \\
& \mathbb{E}_c[t - \sum_{k=1}^{K} p_k^*(c; z, t)] \geq 0.
\end{aligned}
\tag{5}
$$

The second constraint ensures the participation of the data broker in the market, serving as the broker's IR condition. Moreover, the corresponding social welfare is given by $SW_5 = \mathbb{E}_c[-\sum_{k=1}^{K} c_k(x_k^*(c; z, t))^{\rho_k}]$.

A natural question is whether optimal mechanisms for both $(z, t)$ and $(x, p)$ can be derived in polynomial time. The following Algorithm 3 and Theorem 3.1 establish their optimality and computational complexity.

---

**Algorithm 3** Mechanism $\mathcal{M}_5$ for Type 1 platforms with data broker in the three-layer market.

---

*# First-stage mechanism design.*
Replace virtual cost: $vc(\widehat{c})$ by $b$.
Call Algorithm 1 for $y$.
Calculate the allocations: $z_k = y_k^{1/\rho_k}$ for any $k \in [K]$.
**Output:** Allocation-payment pair $(z, t) = ((z_1, ..., z_K), b \cdot y)$.
*# Second-stage mechanism design.*
**Input:** Allocation $z$, payment $t$ and report $\widehat{c}$.
Calculate the payment: $p_k(\widehat{c}) = b_k z_k^{\rho_k}$.
**Output:** Allocation-payment pair $(x(\widehat{c}), p(\widehat{c})) = (z, (b_1 z_1^{\rho_1}, ..., b_K z_K^{\rho_K}))$.

---

**Theorem 3.1.** *Mechanism $\mathcal{M}_5$ is a valid optimal solution to Problems 4 and 5 and achieves the highest expected revenue/lowest cost for the platform with polynomial time complexity.*

### 3.2 Welfare-Maximizing Procurement Mechanism $\mathcal{M}_6$ with a Data Broker

We now turn to platforms that aim to maximize social welfare. The first-stage optimization problem is formulated as follows, while the second-stage problem remains the same as Problem 4. The corresponding platform revenue is $Rev_6 = \mathbb{E}_c[-t]$.

$$
\begin{aligned}
SW_6 = \max_{z,t} \quad & \mathbb{E}_c[-\sum_{k=1}^{K} c_k(x_k^*(c; z, t))^{\rho_k}] \\
\text{s.t.} \quad & z_k + \sum_{i=1}^{K} \mu_{ik} z_i^{\gamma_{ik}} \geq d_k \text{ and } z_k \geq 0 \text{ for any } k \in [K] \\
& \mathbb{E}_c[t - \sum_{k=1}^{K} p_k^*(c; z, t)] \geq 0.
\end{aligned}
\tag{6}
$$

Since the platform must determine $z$ prior to the revelation of $c$, the optimal solution is expected to depend on $\mathbb{E}[c]$. Formally, we substitute $b$ in Algorithm 3 with $\mathbb{E}[c]$ and derive the mechanism $\mathcal{M}_6$, presented in Algorithm 5 of Appendix B, and Theorem 3.2

**Theorem 3.2.** *Mechanism $\mathcal{M}_6$ is a valid optimal solution to Problems 4 and 6 and achieves the highest social welfare with polynomial time complexity.*

### 3.3 The *Lose-Lose* Effect in Three-Layer Markets

Lastly, we compare the platform's revenue and social welfare in the three-layer market with those in the two-layer setting. We show that both revenue and social welfare decline relative to $\mathcal{M}_2$ and $\mathcal{M}_4$, indicating that the presence of data brokers distorts allocations and reduces market efficiency. Public contracting further hinders the effective incentivization of downstream creators and prevents the platform from optimizing social welfare based on realized types. This finding highlights the growing need for government oversight and regulation of data markets (Brooks, 2024).

**Theorem 3.3.** *Compared with $\mathcal{M}_2$ and $\mathcal{M}_4$, both $\mathcal{M}_5$ and $\mathcal{M}_6$ face a lose-lose situation that $Rev_2 \geq Rev_4 = Rev_5 \geq Rev_6$ and $SW_2 = SW_4 \geq SW_6 \geq SW_5$.*

An interesting result in Theorem 3.3 is that $Rev_4$ equals $Rev_5$. Recall that $\mathcal{M}_4$ corresponds to a platform maximizing social welfare, while $\mathcal{M}_5$ represents a revenue-maximizing platform. The loss of market power due to the creator union and the welfare-oriented objective is economically equivalent to the reduction in price discrimination ability resulting from pre-signed contracts. This phenomenon, observed for the first time, offers new insights into procurement mechanism design in the era of GenAI. From Theorem 3.3, we also observe that the introduction of a data broker reduces both platform revenue and social welfare. More surprisingly, the revenue of a Type 1 platform in the three-layer market falls below that of a Type 2 platform in the two-layer market. Moreover, under mechanisms $\mathcal{M}_5$ and $\mathcal{M}_6$, the data broker's profit is exactly zero, indicating no benefit even to himself. This outcome arises because, in the three-layer market, the platform must make decisions before observing creators' cost reports, leading to conservative and distorted allocations to satisfy hard demand constraints. These findings underscore the need for stronger government regulation of third-party data platforms to improve overall social welfare.

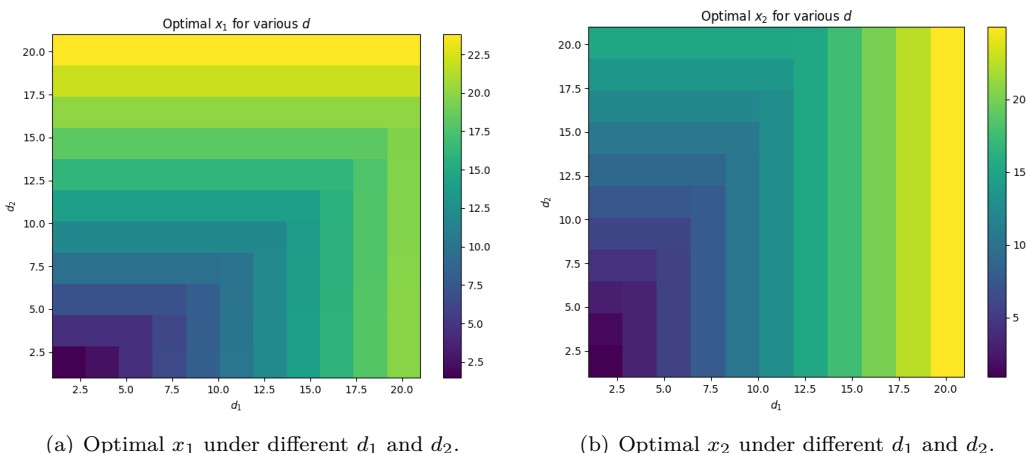

(a) Optimal $x_1$ under different $d_1$ and $d_2$.

(b) Optimal $x_2$ under different $d_1$ and $d_2$.

Figure 2: The overproduction phenomenon with respect to demand $d$.

Finally, we present the following byproduct as a lower bound on the revenue of $\mathcal{M}_3$, based on $\mathcal{M}_5$ and obtained by constructing a feasible solution to Problem 3.

**Corollary 3.1.** *It holds that $Rev_3 \geq Rev_5$, providing a lower bound of $Rev_3$.*

We end this section by noting that although the proof of the lose-lose effect relies on the specific model, we conjecture that it is a pervasive phenomenon in three-layer markets. The fundamental reason for the lose-lose effect is as follows. When a platform and a broker sign a public contract for content procurement, such as the agreement between OpenAI and The Wall Street Journal, such large-scale deals are typically difficult to keep private due to regulatory requirements. As a result, creators in the relevant domain anticipate higher demand for their content and raise their payment requirements. This weakens the platform's ability to price discriminate among human creators and naturally reduces the platform's revenue. For social welfare, once human creators become aware of the demand specified in the broker's contract, the broker must not only compensate them for their production costs but also incentivize them to generate at least a target level of content. This requires the broker to provide additional informational rents to ensure participation and effort. Since social welfare depends solely on the true production costs, these additional incentives introduce distortions in the allocation. As a result, the maximization of social welfare is distorted, leading to lower social welfare in the three-layer market compared to the two-layer market.

## 4 Empirical Studies

**Impact of costs on overproduction.** We conduct synthetic experiments to examine the phenomenon of overproduction under the setting of Section 2.2. For visualization, we consider $K = 2$ domains. The hyperparameters $c$, $\mu$, and $\gamma$ are drawn from Unif$[0, 1]$, while $\rho$ follows Unif$[1, 2]$. All hyperparameters are vector-valued, and subscripts are omitted for clarity. We vary $d_1$ and $d_2$ between 1 and 20 and plot their relationship with the optimal allocations $x_1$ and $x_2$ derived from Algorithm 1. As shown in Figure 2(a), when $d_2$ is high (e.g., between 15 and 20), $x_1$ remains nearly constant regardless of $d_1$, indicating that the constraint $x_1 + \sum_{i=1}^{2} \mu_{i1} x_i^{\gamma_{i1}} \geq d_1$ is non-binding and overproduction occurs in the first domain. The realized costs in this case are 0.035 and 0.992, respectively. Similarly, Figure 2(b) shows overproduction in the second domain when $d_1 \geq 12.5$, where $x_2$ changes little as $d_2$ increases, implying that $x_2 + \sum_{i=1}^{2} \mu_{i2} x_i^{\gamma_{i2}} \geq d_2$ becomes loose. The corresponding realized costs are 0.989 and 0.055. When relative production costs are low, overproduction is more likely to occur. The economic intuition lies in the transferability and substitution effects across domains: producing low-cost content in one domain can help satisfy demand in others through data transfer. Although this leads to excessive production and additional expenditure in the low-cost domain, the resulting savings in high-cost domains can more than offset it. Hence, overproduction in some domains

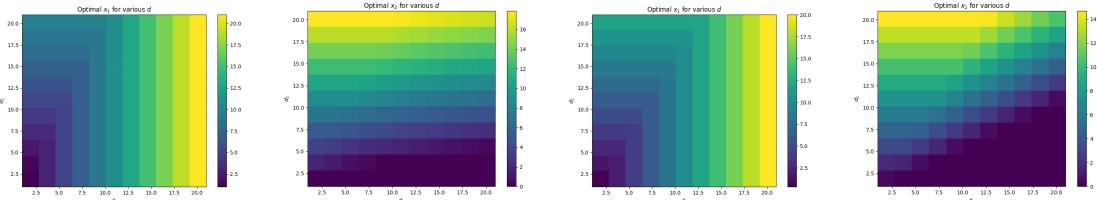

(a) Optimal $x_1$ under differ- (b) Optimal $x_2$ under differ- (c) Optimal $x_1$ under differ- (d) Optimal $x_2$ under different $d_1$ and $d_2$ when $\mu_{12} = 1$. ent $d_1$ and $d_2$ when $\mu_{12} = 1$. ent $d_1$ and $d_2$ when $\gamma_{12} = 1$. ent $d_1$ and $d_2$ when $\gamma_{12} = 1$.

Figure 3: The influence of content transferability on the optimal production.

can ultimately increase platform revenue. Moreover, these experiments empirically support Corollary 2.2, confirming that the optimal allocation $x_k(c)$ remains positive for all cost realizations $c$.

**Impact of transferability parameters on overproduction.** We next examine the specific effects of the transferability parameters $\mu$ and $\gamma$. To isolate their influence, we fix $c_1 = c_2 = 0.5$, $d_1 = d_2 = 10$, and $\rho_1 = \rho_2 = 4$. We first study the role of $\mu$ by setting $\gamma_{11} = \gamma_{12} = \gamma_{21} = \gamma_{22} = 0.5$. Consider an extreme case where $\mu_{12} = 1$ and $\mu_{11} = \mu_{21} = \mu_{22} = 0$, meaning that only content in domain 1 can facilitate production in domain 2, but not vice versa. As shown in Figure 3(a), when $d_2$ is high and $d_1$ is low, overproduction occurs in domain 1 indicated by the uniform color in the upper-left region. This happens because domain 1 content has higher transferability, enabling the platform to train GenAI models that generate outputs in both domains. For instance, when $d_2 = 20$ and $d_1 \leq 10$, the optimal $x_1$ is around 10, well above the demand $d_1$, so the constraint $x_1 + \sum_{i=1}^{2} \mu_{i1} x_i^{\gamma_{i1}} \geq d_1$ is non-binding. As $d_1$ increases, this constraint becomes binding and $x_1$ rises accordingly. In contrast, Figure 3(b) shows no such pattern, since the constraint related to $d_2$ remains binding across all parameter combinations. Interestingly, as $d_1$ increases, the optimal $x_2$ decreases because a larger $x_1$ transfers more knowledge to domain 2, reducing the need for direct content production there. These results highlight the asymmetric effect of $\mu$. We then turn to the influence of $\gamma$ by fixing $\mu_{11} = \mu_{12} = \mu_{21} = \mu_{22} = 0.5$ and setting $\gamma_{12} = 1$, while $\gamma_{11} = \gamma_{21} = \gamma_{22} = 0$. In this case, domain 1 content can continuously support the production of domain 2 content, while the reverse effect remains constant. We again observe overproduction in domain 1. Moreover, when $d_1$ is large but $d_2$ is small, human-created content in domain 1 provides sufficient transferable knowledge, keeping $x_2$ nearly constant and close to zero (Figure 3(d)). Overall, these experiments confirm that higher transferability, whether through $\mu$ or $\gamma$, increases the likelihood of overproduction across domains.

**Experimental evaluation of revenue and welfare among mechanisms.** We next conducted a series of experiments to compare the performance of Algorithms 1 to 3, 4 and 5. The results corroborate Theorems 2.4 and 3.3, showing that the empirical rankings of revenue and social welfare align with the theoretical predictions. As illustrated in Figure 4, mechanism $\mathcal{M}_1$ exhibits more outliers and extreme values compared with $\mathcal{M}_2$. This suggests that welfare-maximizing platforms may operate in a more stable market environment, where both social welfare and revenue fluctuate less, offering a possible explanation for why some companies choose to prioritize social welfare. Detailed results are provided in Appendix E.

**With and Without GenAI.** Finally, we simulate the difference between scenarios with and without GenAI, that is, with $\mu = 0$. Using the same setup as the preceding evaluation and mechanism $\mathcal{M}_1$ as an illustrative example, we report detailed results in Table 2 of Appendix E. In our framework, GenAI leverages human-created content to generate additional material, allowing platforms to meet total demand with less direct human creation. Consequently, both revenue and social welfare increase in the presence of GenAI. Specifically, the sample-averaged revenue rises from –30.90 to –16.52, and sample average social welfare improves from –5.66 to –4.47, representing gains of 46.54% and 21.02%, respectively. These results indicate that GenAI enhances production efficiency while increasing both platform revenue and social welfare. However, as the primary beneficiary of GenAI, the platform enjoys a substantially larger increase in revenue than the corresponding improvement in social welfare. Although the magnitude of improvement depends

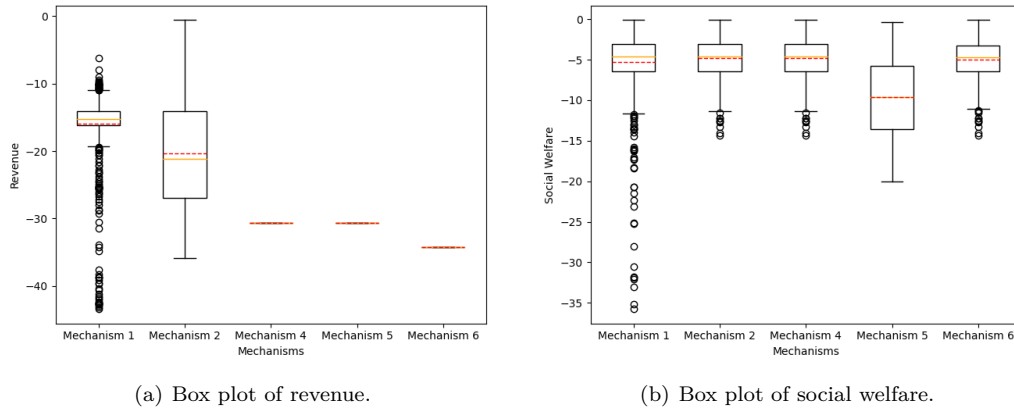

(a) Box plot of revenue.          (b) Box plot of social welfare.

Figure 4: Concrete revenue and social welfare of different mechanisms (red: mean; orange: median).

on parameter initialization, the asymmetric growth between revenue and welfare reflects a broader pattern observed in practice (Hosseini & Khanna, 2025), highlighting reflections regarding the distributional impact of GenAI adoption.

## 5 Conclusion and Discussion

In this paper, we study *non-linear procurement mechanism design* in markets where *human creators coexist with GenAI*. We develop optimal mechanisms for platforms seeking to maximize either revenue or social welfare across three market structures: two-layer markets (with or without a creator union) and three-layer markets involving data brokers. Our analysis shows that although the rapid advancement of GenAI substantially affects creators in certain domains, it cannot fully replace them in competitive environments (Chiang & Lee, 2023). Finally, we uncover a striking result: the three-layer market structure leads to a *lose-lose* outcome, echoing growing societal concerns over AI regulation and governance (Smuha, 2021; de Almeida et al., 2021; Minssen et al., 2023).

This work leaves open many interesting future directions. For general high-dimensional mechanism design problems, such as Mechanism $\mathcal{M}_3$, can the optimal solution be computed exactly in polynomial time? Given that our optimization problem retains partial linearity, could approximate solvers be effective when parameters $\rho$ and $\gamma$ are unknown? Since the three-layer market leads to simultaneous declines in both revenue and social welfare, how should government intervention be structured, for instance, what constitutes an appropriate tax rate for data brokers? Furthermore, does competition among human creators yield excess profits for data intermediaries? Finally, our static model leads to predictions and insights about the ultimate equilibrium outcomes. How will these insights change in dynamic settings when GenAI and human creators dynamically interact? In light of the growing demand for data in the AI era and the distinctive nature of data as an economic good, these questions open several promising directions for future exploration.

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

# A    Omitted Details in Section 1

## A.1    Related Works

We summarize the following four lines of existing literature pertinent to our work.

**Procurement Mechanism Design.**    The topic of optimal mechanism design was introduced by the seminal paper Myerson (1981). Over decades of academic attention, researchers have not only focused on optimal mechanism design from the seller's perspective but have also begun to consider it from the buyer's perspective, especially when buyers are either significantly large or even monopolistic, leading to studies on optimal procurement mechanism design (McAfee & McMillan, 1987; Laffont & Tirole, 1993). In the economics literature, Burguet et al. (2012) examines mechanism design in procurement under limited liability and explores ways to reduce the sponsor's cost, while Gerding et al. (2010) addresses the service procurement problem in the presence of uncertain duration. Garg & Narahari (2008) adopts the framework of a Stackelberg game to study procurement auction design. In management science and supply chain literature, Zhang (2010) investigates procurement with price-sensitive demand, while Prasad & Rao (2013); Ketankumar et al. (2015) focus on the procurement of cloud computing resources. Jin & Wu (2002); Huang et al. (2011) attempt to design optimal procurement auctions to enhance suppliers' profit margins. Our paper goes beyond traditional procurement mechanism design for physical goods by focusing on the digital content market, which features transferability. This extends the boundaries of procurement mechanism design in this domain.

**GenAI Content Creation and Data Pricing.**    The development of generative AI has brought AI-generated content into the economic market, sparking research on the nature of AI products and traditional goods. Liu et al. (2024a) discusses the impact of large language models (LLM) on traditional search engines, while Zhang et al. (2024b) explores advertising business in the context of LLMs. Iyer et al. (2024) conducts a case study on the role of LLMs in e-commerce. Chkirbene et al. (2024) provides a comprehensive survey of GenAI's applications, challenges, and trends from several aspects, including content creation and marketing. Meanwhile, due to the enormous data demand for GenAI training, public datasets are becoming insufficient, leading companies to shift toward private datasets, which has raised concerns about data pricing. Data Shapley and its variants are the most commonly used methods for data pricing (Jia et al., 2019; Ghorbani & Zou, 2019; Schoch et al., 2022), while recent Ai et al. (2024) proposes a novel data pricing method based on instrumental value. Our paper studies how much payment is needed to incentivize human creators to generate sufficient content within an agent-based framework, broadening the perspective on data pricing related to GenAI.

**Human vs. Generative AI Competition.**    With the rapid development of generative AI, competition between human creators and GenAI has gradually emerged (Esmaeili et al., 2024). In contrast to traditional automation (Acemoglu & Restrepo, 2019), GenAI relies on human-created data for training, thereby simultaneously complementing and competing with human labor. Yao et al. (2024) employs the framework of a Tullock contest to examine the symbiosis and conflict between humans and generative AI. Immorlica et al. (2024) examines the augmentation of users by AI agents and the corresponding shifts in equilibrium. Gao et al. (2025), within a game-theoretic framework, investigates the impact of AI tools on market outcomes and consumer welfare. It highlights that in certain scenarios, both platforms and creators may be reluctant to adopt GenAI. Moreover, some papers discuss the unique economic characteristics of LLMs, or broadly GenAI; please refer to Fish et al. (2024); Duetting et al. (2024) for further details. Meanwhile, some papers explore the collaboration between humans and GenAI in various tasks (Fui-Hoon Nah et al., 2023; Singh et al., 2023; Han et al., 2024). We adopt a mechanism design perspective to study the collaboration and competition between human creators and GenAI. GenAI relies on human-created content for training, while the content generated by AI partially substitutes human-created content, providing a novel approach to studying the competition between humans and AI.

**Three-Layer Markets.**    The three-layer market offers a more comprehensive framework for characterizing data markets compared to the two-layer market. For further details, Zhang et al. (2024a) provides a survey on data markets. Fallah et al. (2024) models the three-layer data market as a multistage game and focuses

on the subgame Nash equilibrium, while Xu et al. (2020) considers a three-layer Stackelberg game in the car-sharing market. Besides, Reinders et al. (2018); Song & He (2019) consider three-layer structures in the domain of supply chain. Wang et al. (2023) investigates privacy protection issues in a blockchain-based P2P three-layer market. Balseiro et al. (2021); Zeithammer & Choi (2024) investigate the impact of intermediaries on the payoffs of buyers and sellers in auctions, as well as the phenomenon of double-shading in the market. We compare the impacts of two-layer and three-layer markets, within the framework of competition between humans and GenAI, on both platform revenue and social welfare. We theoretically demonstrate that the three-layer structure would lead to a lose-lose outcome, offering a novel perspective for future AI data regulation.

### A.2 Relative Terms

We use this subsection to illustrate some terminology used in this paper for readers with less economic background. These terms are definitional, but we also provide the economic intuition behind them to assist the reader's understanding.

- Virtual costs: A variant of the original cost, formally defined as $c + \frac{F}{f}$. Because the platform must incentivize creators to report their true costs, the effective cost the platform faces is the human creator's production cost $c$ plus the incentive-related virtual cost $\frac{F}{f}$.

- Reverse hazard rates: Originating from survival analysis (Kalbfleisch & Prentice, 2002), and defined in this paper as $\frac{f}{F}$. It is used to characterize the log-concavity of $F$.

- Incentive compatibility: The terminology for a desirable mechanism property that for any creator with private cost $c$, truthfully reporting $c$ to the platform always maximizes the creator's expected utility (payment minus cost). Hence incentive incentive-compatible mechanisms can help avoid dishonest agent behaviors during information elicitation.

- Individual rationality: The terminology of another desirable mechanism property that human creators always have non-negative expected utility (so they prefer to stay in the market than to exit).

- Path independence: When payments are defined through a path integral (e.g., $\int_c^b y(t) \cdot dt$ in Mechanism $\mathcal{M}_4$), we require that the integral yields the same value for any path from $\hat{c}$ to $b$.

- Myerson's analysis: This refers to a classic and well-known analysis framework of Myerson (1981) that converts payments into integrals of the allocation functions, which reduces mechanism design to allocation rule design.

- Convex procurement: We hope the feasible set induced by demand constraints to be convex, so that the optimal mechanism can be computed efficiently in practice.

- Valid mechanisms: A creator's higher cost should lead the platform to procure less from them, while other creators' higher costs should lead the platform to procure more from this creator. This reflects substitutability in production.

## B Omitted Optimal Mechanisms and Algorithms

We now detail the concrete implementation of mechanisms $\mathcal{M}_2$ and $\mathcal{M}_6$.

## C Omitted Details in Section 2

### C.1 Including GenAI Costs

In practice, the generation cost of GenAI is typically negligible compared to that of human creators. For example, once a large model such as Sora 2 is already trained, producing a new video takes only a few

---

**Algorithm 4** Mechanism $\mathcal{M}_2$ for welfare-maximizing platforms without union in the two-layer market.

---

**Input:** Report $\widehat{c}$.
Replace virtual cost: $vc(\widehat{c})$ by $\widehat{c}$.
Call Algorithm 1 for $y$.
Calculate the allocations: $x_k(\widehat{c}) = y_k^{1/\rho_k}$ for any $k \in [K]$.
Calculate the payments: $p_k(\widehat{c}) = \widehat{c}_k y_k + \int_{c_k}^{b_k} y_k(\widehat{c}_1, ..., \widehat{c}_{k-1}, t_k, \widehat{c}_{k+1}, ..., \widehat{c}_K) dt_k$ for any $k \in [K]$.
**Output:** Allocation-payment pair $(x(\widehat{c}), p(\widehat{c})) = ((x_1(\widehat{c}), ..., x_K(\widehat{c})), (p_1(\widehat{c}), ..., p_K(\widehat{c})))$.

---

**Algorithm 5** Mechanism $\mathcal{M}_6$ for Type 2 platforms with data broker in the three-layer market.

---

*# First-stage mechanism design.*
Replace virtual cost: $vc(\widehat{c})$ by $\mathbb{E}[c]$.
Call Algorithm 1 for $y$.
Calculate the allocations: $z_k = y_k^{1/\rho_k}$ for any $k \in [K]$.
**Output:** Allocation-payment pair $(z, t) = ((z_1, ..., z_K), b \cdot y)$.
*# Second-stage mechanism design.*
**Input:** Allocation $z$, payment $t$ and report $\widehat{c}$.
Calculate the payment: $p_k(\widehat{c}) = b_k z_k^{\rho_k}$.
**Output:** Allocation-payment pair $(x(\widehat{c}), p(\widehat{c})) = (z, (b_1 z_1^{\rho_1}, ..., b_K z_K^{\rho_K}))$.

---

minutes of computation, whereas professional human creators require substantial compensation and effort to produce comparable content.

However, our model can naturally extend to settings with small but non-zero GenAI training costs. We assume the training cost of GenAI is proportional to human production cost, and the training cost parameter of GenAI is $c_k^G$ in domain $k$. In other words, the platform needs to pay $c_k^G x_k^{\rho_k}$ to obtain a GenAI agent for domain $k$. Consider Mechanism $\mathcal{M}_1$ as an example: our transformation is derived entirely from IC and IR constraints, independent of the objective function. Therefore, incorporating GenAI costs is equivalent to adding $\sum_{k=1}^{K} c_k^G y_k$ to the platform's objective. The optimization problem in Algorithm 1 becomes as follows:

$$y = \underset{y}{\operatorname{argmin}} \sum_{k=1}^{K} (vc_k(\widehat{c}) + c_k^G) y_k$$

$$\text{s.t. } y_k^{1/\rho_k} + \sum_{i=1}^{K} \mu_{ik} y_i^{\gamma_{ik}/\rho_i} \geq d_k \text{ and } y_k \geq 0 \text{ for any } k \in [K].$$

As a result, we only need to adjust the virtual cost from $vc_k$ to $vc_k + c_k^G$ to recover the optimal mechanism under non-zero GenAI costs.

The same modification applies to other mechanisms as well. For instance, in Mechanism $\mathcal{M}_2$, replacing $\widehat{c}_k$ with $\widehat{c}_k + c_k^G$ leaves the structure and insights unchanged. For other kinds of GenAI costs, we can similarly include the cost in the objective without modifying the demand constraints, with or without access to a closed-form solution, depending on the concrete cost form.

## C.2 Further Complexity Reduction

Note that we adopt a deterministic payment rule in Algorithm 1. However, we can move to a random manner as follows to reduce the computational complexity in Proposition 2.1. Here, $y_k(\widehat{c}_1, ..., \widehat{c}_{k-1}, \widetilde{c}_k, \widehat{c}_{k+1}, \widehat{c}_K)$ indicates that the input report is $(\widehat{c}_1, ..., \widehat{c}_{k-1}, \widetilde{c}_k, \widehat{c}_{k+1}, \widehat{c}_K)$ without ambiguity.

$$\text{Sample } \widetilde{c}_k \sim \text{Unif}(\widehat{c}_k, b_k) \text{ and } p_k(\widehat{c}) = \widehat{c}_k y_k(\widehat{c}) + (b_k - \widehat{c}_k) y_k(\widehat{c}_1, ..., \widehat{c}_{k-1}, \widetilde{c}_k, \widehat{c}_{k+1}, \widehat{c}_K). \tag{7}$$

**Proposition C.1.** *Replacing the payment rule in Algorithm 1 by Equation* (7)*, using the subgradient method as the oracle, the computational complexity of getting a $\epsilon$-optimal mechanism is $\mathcal{O}(\frac{1}{\epsilon^2})$. Specially, if $\max_i\{\frac{\gamma_{ik}}{\rho_i}\} < 1$ for any $k \in [K]$, the computational complexity reduces to $\mathcal{O}(\frac{1}{\epsilon})$.*

### C.3 Omitted Proof in Section 2.2

### C.3.1 Proof of Theorem 2.1

Recall Problem 1 and we use $y_k(c) = x_k^{\rho_k}(c)$ to simplify the notations. Besides, we use $Q_k(y, c_k)$ and $u_k(y, p, c_k)$ to denote $\int y_k(c) f_{-k}(c_{-k}) dc_{-k}$ and $\int [p_k(c) - c_k y_k(c)] f_{-k}(c_{-k}) dc_{-k}$ respectively. Therefore, $Q$ represents the units human creators will produce and $u$ represents their expected utility.

Then, we have the following propositions regarding $Q.(\cdot, \cdot)$ and $u.(\cdot, \cdot, \cdot)$.

**Proposition C.2.** *We have that*

- *for any $c_k \in [a_k, b_k]$, it holds that $u_k(y, p, c_k) = u_k(y, p, b_k) + \int_{c_k}^{b_k} Q_k(y, t_k) dt_k$.*

- *for any $k \in [K]$, it holds that $u_k(y, p, b_k) \geq 0$.*

- *for any $t_k \geq c_k$, it holds that $Q_k(y, t_k) \leq Q_k(y, c_k)$.*

*Proof.* From the individual rationality condition, we know that the expected utility for human creators in domain $k$ at every time is no less than zero. Then, it holds especially when the cost corresponds to $b_k$. It immediately leads to $u_k(y, p, b_k) \geq 0$.

Assuming the real cost parameter is $t_k$ but the human report $c_k$. Assume $c_k \leq t_k$ without loss of generality. From the incentive compatibility condition, it holds that

$$
\begin{aligned}
u_k(y, p, t_k) &\geq \int [p_k(c_k) - t_k y_k(c)] f_{-k}(t_{-k}) dt_{-k} \\
&= \int [p_k(c_k) - c_k y_k(c)] f_{-k}(t_{-k}) dt_{-k} + (c_k - t_k) \int y_k(c) f_{-k}(t_{-k}) dt_{-k} \\
&= u_k(y, p, c_k) + (c_k - t_k) Q_k(y, c_k).
\end{aligned}
$$

Then, we know that $u_k(y, p, t_k) - u_k(y, p, c_k) \geq (c_k - t_k) Q_k(y, c_k)$.

Similarly, we have that $u_k(y, p, t_k) - u_k(y, p, c_k) \leq (c_k - t_k) Q_k(y, t_k)$ by changing the order of $t_k$ and $c_k$. Thus, it holds that

$$
(c_k - t_k) Q_k(y, c_k) \leq u_k(y, p, t_k) - u_k(y, p, c_k) \leq (c_k - t_k) Q_k(y, t_k).
$$

Since $c_k \leq t_k$, we know that $Q_k(y, t_k) \leq Q_k(y, c_k)$.

In addition, dividing $c_k - t_k$, it holds that $\frac{\partial u_k}{\partial c_k} = -Q_k(y, c_k)$. Here we use the fact that $Q_k(y, \cdot)$ is continuous which will be shown soon. Otherwise, we can still get the following from the definition of Riemann integral. We then obtain that $u_k(y, p, c_k) + \int_{c_k}^{t_k} -Q_k(y, t_k) dt_k = u_k(y, p, t_k)$. Therefore, with simple algebra, we have that

$$
u_k(y, p, c_k) = u_k(y, p, b_k) + \int_{c_k}^{b_k} Q_k(y, t_k) dt_k,
$$

which ends the proof. □

Now, let's turn to the problem of how to maximize content creation platforms' expected revenue (minimize expected cost). We use $u_0(y, p) = \sum_{k=1}^{K} \int -p_k(c) f(c) dc$ to denote the expected revenue for shorthand. Recall that the demand constraints are now

$$
y_k^{1/\rho_k}(c) + \sum_{i=1}^{K} \mu_{ik} y_i(c)^{\gamma_{ik}/\rho_i} \geq d_k
$$

and

$$
y_k(c) \geq 0
$$

for any $k \in [K]$.

We first decompose $u_0(y, p)$ as follows, namely,

$$u_0(y, p) = \underbrace{\sum_{k=1}^{K} \int [-p_k(c) + c_k y_k(c)] f(c) dc}_{q_1} - \underbrace{\sum_{k=1}^{K} \int c_k y_k(c) f(c) dc}_{q_2}.$$

Then, it holds that

$$q_1 = -\sum_k u_k(y, p, c_k) f_k(c_k) dc_k$$

$$= -\sum_k \int [u_k(y, p, b_k) + \int_{c_k}^{b_k} Q_k(y, t_k) dt_k] f_k(c_k) dc_k$$

$$= \sum_k -u_k(y, p, b_k) - \int_{a_k}^{b_k} \int_{c_k}^{b_k} f_k(c_k) Q_k(y, t_k) dt_k dc_k$$

$$= \sum_k -u_k(y, p, b_k) - \int_{a_k}^{b_k} F_k(t_k) Q_k(y, t_k) dt_k$$

$$= \sum_k -u_k(y, p, b_k) - \int F_k(t_k) \int y_k(t) f_{-k}(t_{-k}) dt$$

$$= \sum_k -u_k(y, p, b_k) - \int \frac{F_k(t_k)}{f_k(t_k)} y_k(t) f(t) dt.$$

The third equation holds due to the property of p.d.f. while the fourth equation holds because we change the order of integration.

Combining $q_2$, it holds that

$$u_0(y, p) = \sum_{k=1}^{K} -u_k(y, p, b_k) - \int [c_k + \frac{F_k(c_k)}{f_k(c_k)}] y_k(c) f(c) dc.$$

By defining the virtual cost $vc_k(c) = c_k + \frac{F_k(c_k)}{f_k(c_k)}$, then we have

$$u_0(y, p) = \sum_{k=1}^{K} -u_k(y, p, b_k) - \int vc_k(c_k) y_k(c) f(c) dc.$$

Note that in Algorithm 1, we minimize $\sum_{k=1}^{K} vc_k(c_k) y_k(c)$ subject to $y_k^{1/\rho_k}(c) + \sum_{i=1}^{K} \mu_{ik} y_i(c)^{\gamma_{ik}/\rho_i} \geq d_k$ and $y_k(c) \geq 0$ for any $k \in [K]$, so we only need to prove that the payment rule indeed results in $u_k(y, p, b_k) = 0$. Here, since we consider an IC mechanism, we have naturally $c = \hat{c}$.

Recall that from Proposition C.2, it holds that

$$u_k(y, p, b_k) = u_k(y, p, c_k) - \int_{c_k}^{b_k} Q_k(y, t_k) dt_k$$

$$= -\int [c_k y_k(c) - p_k(c)] f_{-k}(c_{-k}) dc_{-k} - \int_{c_k}^{b_k} \int y_k(t_k, c_{-k}) f_{-k}(c_{-k}) dc_{-k} dt_k$$

$$= \int [-c_k y_k(c) + p_k(c) - \int_{c_k}^{b_k} y_k(t_k, c_{-k}) dt_k] f_{-k}(c_{-k}) dc_{-k}.$$

Here, we use $(t_k, c_{-k})$ to represent $(c_1, ..., c_{k-1}, t_k, c_{t+1}, ..., c_K)$ with a little abuse of notation. Since we set the payment rule $p_k(c) = c_k y_k(c) + \int_{c_k}^{b_k} y_k(t_k, c_{-k}) dt_k$ in Algorithm 1, we know that $u_k(y, p, b_k) = 0$ exactly.

Together with Proposition C.2, we know that the first term in $u_0(y, p)$ is maximized as well. Therefore, since two parts in $u_0(y, p)$ are maximized simultaneously, $u_0(y, p)$ takes its maximum using Algorithm 1, which shows the optimality of our mechanism.

Additionally, since the allocation-related $y_k(\cdot, \cdot) \geq 0$, we know that

$$-c_k y_k(c) + p_k(c) = \int_{c_k}^{b_k} y_k(c_1, ..., c_{k-1}, t_k, c_{k+1}, ..., c_K) dt_k \geq 0$$

for every $c$. Thus, we actually obtain a much stronger individual rationality condition, say $-c_k x_k^{\rho_k}(c) + p_k(c) \geq 0$. In other words, the utility of human creators is non-negative for any $k \in [K]$ and cost vector $c \in [a_1, b_1] \times \cdots \times [a_k, b_k]$. So, of course, the individual rationality holds in the expectation manner, and human creators won't exit the market due to a negative utility.

As for the incentive compatibility condition, we first show that $vc_k(c)$ is increasing with respect to $c_k$. Since we have Assumption 2.3, we know that $\log(F_k(\cdot))$ is concave, it holds that $\frac{f_k}{F_k}$ is decreasing. Therefore, it holds that $vc_k(c) = c_k + \frac{F_k(c_k)}{f_k(c_k)}$ is increasing in $c_k$ for all $k \in [K]$. Together with the implementation of Algorithm 1, we know the objective function is linear in $y$, and from the following Lemma C.3 we will see the feasible region is convex, then we can actually derive similarly a stronger version of the IC condition, namely, $-c_k x_k^{\rho_k}(c) + p_k(c) \geq -c_k x_k^{\rho_k}(\widehat{c}_k, c_{-k}) + p_k(\widehat{c}_k, c_{-k})$. Specifically, when the true cost is $c_k$ but the human creator reports $\widehat{c}_k$, it holds that the creator will have a utility loss

$$(c_k - \widehat{c}_k) y_k(c_1, ..., c_{k-1}, \widehat{c}_k, c_{k+1}, ..., c_K) + \int_{c_k}^{\widehat{c}_k} y_k(c_1, ..., c_{k-1}, t_k, c_{k+1}, ..., c_K) dt_k.$$

For overreport $\widehat{c}_k > c_k$, it holds that $y_k(c_1, ..., c_{k-1}, \widehat{c}_k, c_{k+1}, ..., c_K) \leq y_k(c_1, ..., c_{k-1}, t_k, c_{k+1}, ..., c_K)$ for every $t_k \in [c_k, \widehat{c}_k]$ in our convex optimization problem. Meanwhile, for underreport $\widehat{c}_k < c_k$, we know that $y_k(c_1, ..., c_{k-1}, \widehat{c}_k, c_{k+1}, ..., c_K) \geq y_k(c_1, ..., c_{k-1}, t_k, c_{k+1}, ..., c_K)$ for every $t_k \in [\widehat{c}_k, c_k]$. In both cases, the agent incurs a non-negative loss. Therefore, truthful reporting is a dominant strategy, which implies incentive compatibility. Similarly, we get that $Q_k(y, \cdot)$ is continuous almost everywhere, and $\mathcal{M}_1$ is valid for the same reason.

### C.3.2 Proof of Proposition 2.1

We first prove the convexity of the area and then give the computational complexity.

**Lemma C.3.** $y_k^{1/\rho_k} + \sum_{i=1}^K \mu_{ik} y_i^{\gamma_{ik}/\rho_i} \geq d_k$ and $0 \leq y_k \leq d_k^{\rho_k}$ for any $k \in [K]$ construct a convex set. Specially, when $\max_i\{\frac{\gamma_{ik}}{\rho_i}\} < 1$, it becomes a strongly convex set.

*Proof.* Assuming $f_k(y) = y_k^{1/\rho_k} + \sum_{i=1}^K \mu_{ik} y_i^{\gamma_{ik}/\rho_i}$, it holds that

$$\nabla^2 f_k(y) = \text{diag}_i\{\mu_{ik}\frac{\gamma_{ik}}{\rho_i}(\frac{\gamma_{ik}}{\rho_i} - 1)y_i^{\frac{\gamma_{ik}}{\rho_i} - 2} + \frac{1}{\rho_k}(\frac{1}{\rho_k} - 1)y_k^{\frac{1}{\rho_k} - 2}\mathbb{1}\{i = k\}\}.$$

Since we have $\rho_k \geq 1$ and $\gamma_{ik} \leq 1$, it holds that $\nabla^2 f_k(y)$ is a diagonal matrix and all components are non-positive due to $0 \leq y_k \leq d_k^{\rho_k}$. Hence, we know that $f_k(y)$ is a concave function, showing that $f_k(y) \geq d_k$ is a convex set.

When $\max_i\{\frac{\gamma_{ik}}{\rho_i}\} < 1$, it holds that all components of $\nabla^2 f_k(y)$ are negative. Therefore, we have that $f_k(x)$ is a strongly concave function, and similarly, $f_k(y) \geq d_k$ yields a strongly convex set.

Note that the intersection of (strongly) convex sets is also a (strongly) convex set. We know that constraints $y_k^{1/\rho_k} + \sum_{i=1}^K \mu_{ik} y_i^{\gamma_{ik}/\rho_i} \geq d_k$ and $0 \leq y_k \leq d_k^{\rho_k}$ for any $k \in [K]$ yield a convex set. In the meanwhile, we know that if $\max_i\{\frac{\gamma_{ik}}{\rho_i}\} < 1$, the set will become a strongly convex set, which ends the proof. $\square$

With Lemma C.3 in hands, we know that the computational complexity of Algorithm 1 equalling finding the minimum of a (strongly) convex function with primal-dual methods. Therefore, using the subgradient

method (Nesterov, 2009; 2014) and constraining $0 \leq y_k \leq d_k^{\rho_k}$ lead to an $\mathcal{O}(\frac{1}{\epsilon^2})$ complexity to calculate one allocation $y$ immediately. When $\max_i\{\frac{\gamma_{ik}}{\rho_i}\} < 1$, the strongly convexity leads to $\mathcal{O}(\frac{1}{\epsilon})$ complexity instantly (Nesterov, 2005).

For the price vector $p(\widehat{c})$, we use the Monte Carlo method to estimate the second term, i.e., $\int_{c_k}^{b_k} y_k(t_k, c_{-k})dt_k$. Since from the implementation of Algorithm 1, we know that $y_k$ is bounded by $d_k^{\rho_k}$. Then, Rubinstein & Kroese (2016) tells us that we need $\mathcal{O}(\frac{1}{\epsilon^2})$ times of simulation to estimate it with tolerance $\epsilon$. Therefore, the total computational complexity is $\mathcal{O}(\frac{1}{\epsilon^4})$ for general cases and $\mathcal{O}(\frac{1}{\epsilon^3})$ when $\max_i\{\frac{\gamma_{ik}}{\rho_i}\} < 1$ respectively.

### C.3.3 Proof of Proposition C.1

Due to the linearity of $p(\cdot)$, we only need to prove that Equation (7) gives an unbiased estimator of the optimal payment rule, i.e., $\mathbb{E}_{\widetilde{c}_k}[(b_k - \widehat{c}_k)y_k(\widehat{c}_1, ..., \widehat{c}_{k-1}, \widetilde{c}_k, \widehat{c}_{k+1}, ..., \widehat{c}_K)] = \int_{\widehat{c}_k}^{b_k} y_k(\widehat{c}_1, ..., \widehat{c}_{k-1}, t_k, \widehat{c}_{k+1}, ..., \widehat{c}_K)dt_k$.

It holds that

$$\mathbb{E}_{\widetilde{c}_k}[(b_k - \widehat{c}_k)y_k(\widehat{c}_1, ..., \widehat{c}_{k-1}, \widetilde{c}_k, \widehat{c}_{k+1}, ..., \widehat{c}_K)] = (b_k - \widehat{c}_k)\int_{\widehat{c}_k}^{b_k} y_k(\widehat{c}_1, ..., \widehat{c}_{k-1}, t_k, \widehat{c}_{k+1}, ..., \widehat{c}_K)\frac{1}{b_k - \widehat{c}_k}dt_k$$

$$= \int_{\widehat{c}_k}^{b_k} y_k(\widehat{c}_1, ..., \widehat{c}_{k-1}, t_k, \widehat{c}_{k+1}, ..., \widehat{c}_K)dt_k,$$

where the first equation holds because $\widetilde{c}_k$ follows an uniform distribution over $[\widehat{c}_k, b_k]$.

Therefore, we know that we give an IC,IR and $\epsilon$-optimal mechanism with only one sample, namely $y_k(\widehat{c}_1, ..., \widehat{c}_{k-1}, \widetilde{c}_k, \widehat{c}_{k+1}, ..., \widehat{c}_K)$. With the proof of Proposition 2.1, we know that the computational complexity is $\mathcal{O}(\frac{1}{\epsilon^2})$ for general cases and $\mathcal{O}(\frac{1}{\epsilon})$ when $\max_i\{\frac{\gamma_{ik}}{\rho_i}\} < 1$ for any $k \in [K]$.

Nonetheless, the variance of the payment increases from $\mathcal{O}(\epsilon^2)$ to $\mathcal{O}(1)$ as we decrease the number of samples when using the Monte Carlo method from $\mathcal{O}(\frac{1}{\epsilon^2})$ to constant. To better balance the tradeoff between computational complexity and variance, i.e., uncertainty, we may need $n$ samples to estimate $\int_{\widehat{c}_k}^{b_k} y_k(\widehat{c}_1, ..., \widehat{c}_{k-1}, t_k, \widehat{c}_{k+1}, ..., \widehat{c}_K)dt_k$, yielding $\mathcal{O}(\frac{1}{n})$ variance and $\mathcal{O}(\frac{n}{\epsilon^2})/\mathcal{O}(\frac{n}{\epsilon})$ complexity.

### C.3.4 Proof of Corollary 2.2

From the constraint $y_k^{1/\rho_k} + \sum_{i=1}^{K} \mu_{ik}y_i^{\gamma_{ik}/\rho_i} \geq d_k$, we know that when $y_k = 0$, the corresponding slope is infinity. Besides, from the knowledge of convex optimization, we know that $(vc_1, ..., vc_K)$ is associated with the corresponding subgradient. However, due to Assumption 2.2, $vc_k$ is upper bounded and finite. Therefore, the solution of $y$ is strictly positive in all components given $\mu_{ik} \neq 0$ and $\gamma_{ik} \neq 0$, which ends the proof.

Since the feasible area is an intersection of $K$ convex sets, when the optimal $y$ is located on the boundary of the $k$-th set, the $k$-th demand constraint will be tight, i.e., binding. Otherwise, it will be loose, i.e., non-binding. Then, we know that although all domains still exist, some will experience overproduction and it completes our analysis.

## C.4 Extension to Multiple Human Creators in Each Domain

We conclude that the extension will retain the convexity of the optimization problem, hence preserving the validity of Theorem 2.1 and Proposition 2.1.

We first write down the revenue-maximizing problem as Problem 1. We assume there are $n_k$ human creators in the $k$-th domain and we use $x_k^i$ ($c_k^i$, $p_k^i$ resp.) for the $i$-th creators where $i \in [n_k]$. We assume these $n_i$ humans have i.i.d. cost function. We stack $c_k^i$ as $c$ and we use $y_k^i$ to denote $(x_k^i)^{\rho_k}$ as before. With these

preparations, we have the following optimization problem,

$$\max_{x,p} \mathbb{E}_c[-\sum_{k=1}^{K}\sum_{i=1}^{n_k} p_k^i(c)]$$

$$\text{s.t. } \sum_{i=1}^{n_k} x_k^i(c) + \sum_{j=1}^{K} \mu_{jk}[\sum_{i=1}^{n_j} x_j^i(c)]^{\gamma_{jk}} \geq d_k \text{ for any } k \in [K]$$

$$\mathbb{E}_{-(k,i)}[-c_k^i(x_k^i(c))^{\rho_k} + p_k^i(c)] \geq \mathbb{E}_{-(k,i)}[-c_k^i(x_k^i(\widehat{c}))^{\rho_k} + p_k^i(\widehat{c})]$$

$$\mathbb{E}_{-(k,i)}[-c_k^i(x_k^i(c))^{\rho_k} + p_k^i(c)] \geq 0$$

$$x_k^i(c) \geq 0 \text{ for any } (k,i) \in [K] \times [n_k],$$

where $(k,i)$ means taking expectation over all other human creators.

Since the IC and IR conditions hold individually, we know the optimization goal becomes $y(c) = \text{argmin}_y \sum_{k=1}^{K}\sum_{i=1}^{n_k}[c_k^i + \frac{F_k(c_k^i)}{f_k(c_k^i)}]y_k^i$ accordingly. We are now ready to prove that the constraints still induce a convex set.

By replacing $(x_k^i)^{\rho_k}$ by $y_k^i$, the last constraint turns to $y_k^i \geq 0$. The first constraint is now $\sum_{i=1}^{n_k}(y_k^i)^{1/\rho_k} + \sum_{j=1}^{K} \mu_{jk}[\sum_{i=1}^{n_j}(y_j^i)^{1/\rho_j}]^{\gamma_{jk}} \geq d_k$. Note that $\rho_k \geq 1$ for any $k \in [K]$ and $\gamma_{jk} \in [0,1]$. We know that $\sum_{i=1}^{n_k}(y_k^i)^{1/\rho_k} + \sum_{j=1}^{K} \mu_{jk}[\sum_{i=1}^{n_j}(y_j^i)^{1/\rho_j}]^{\gamma_{jk}}$ is a concave function with respect to $y$. Therefore, the area corresponding to larger than $d_k$ is a convex set. It soon holds that $y$ belongs to a convex set because the intersection of convex sets is still a convex set.

We conclude that even with multiple human creators in each domain, the revenue-maximizing problem is still a convex optimization problem. We can find the optimal mechanism by solving

$$y(c) = \text{argmin}_y \sum_{k=1}^{K}\sum_{i=1}^{n_k}[c_k^i + \frac{F_k(c_k^i)}{f_k(c_k^i)}]y_k^i$$

$$\text{s.t. } \sum_{i=1}^{n_k}(y_k^i)^{1/\rho_k} + \sum_{j=1}^{K} \mu_{jk}[\sum_{i=1}^{n_j}(y_j^i)^{1/\rho_j}]^{\gamma_{jk}} \geq d_k \text{ for any } k \in [K]$$

$$y_k^i \geq 0 \text{ for any } (k,i) \in [K] \times [n_k].$$

Additionally, the proofs of Theorem 2.1 and proposition 2.1 still hold as they are applicable for general convex optimization problems, so the optimality and the complexity results remain valid. Moreover, we can choose a different $c_k$ for each domain and then allocate assignments among human creators, showing the equivalence of considering only one content creator.

Moreover, we remark that we assume creators in domain $k$ share a common $\rho_k$ to reflect the similar economic characteristics of creators within the same domain. Mathematically, our results do not rely on the homogeneity of $\rho_k$. Suppose that the $i$-th creator in domain $k$ has cost $c_k^i(x_k^i)^{\rho_k^i}$. By defining $y_k^i = (x_k^i)^{\rho_k^i}$, the optimization problem can be rewritten as

$$y(c) = \text{argmin}_y \sum_{k=1}^{K}\sum_{i=1}^{n_k}[c_k^i + \frac{F_k(c_k^i)}{f_k(c_k^i)}]y_k^i$$

$$\text{s.t. } \sum_{i=1}^{n_k}(y_k^i)^{1/\rho_k^i} + \sum_{j=1}^{K} \mu_{jk}[\sum_{i=1}^{n_j}(y_j^i)^{1/\rho_j^i}]^{\gamma_{jk}} \geq d_k \text{ for any } k \in [K]$$

$$y_k^i \geq 0 \text{ for any } (k,i) \in [K] \times [n_k],$$

which still preserves convexity after the transformation. This illustrates the generality of our model.

### C.5 Omitted Proof in Section 2.3

### C.5.1 Proof of Theorem 2.2

From the proof of Theorem 2.1, we know that to satisfy IC, the payment rule has to be $p_k(c) = c_k y_k + \int_{c_k}^{b_k} y_k(c_{-k}, t_k)dt_k + C$ where $c_{-k}$ means all costs except the $k$-th entry and $C$ is a global constant. To guarantee the participation or the IR constraint, we need that $p_k(c) \geq c_k y_k$ because of the parameter transformation $y_k(c) = x_k(c)^{\rho_k}$. From the implementation of Algorithm 1, we know that for any positive virtual cost, $y_k(\cdot)$ is larger than zero letting alone $c$. Thus, it holds that $\int_{c_k}^{b_k} y_k(c_{-k}, t_k)dt_k + C \geq C$. If $C$ is not negative, we know that the IR constraint is satisfied. Considering the corner case that $c = b$, we find that the integral is zero, so we need $C$ to be no smaller than zero. Combining these two results, we know that $C = 0$ is the optimal choice. This shows the feasibility and the optimality of our payment rule.

As we have shown that for every allocation rule, we can find a corresponding payment to motivate its implementation, we know that we only need to set the virtual cost equaling to the true cost $c$ noting that we hope to obtain the lowest social cost, namely $\min_x \sum_k c_k x_k^{\rho_k}(c) = \min_y \sum_k c_k y_k(c)$. It yields the optimality of Algorithm 4 as it satisfies both IC and IR, and achieves the highest social welfare while fulfilling all demands. Since we only need $vc_k(\cdot)$ to be increasing to obtain the validity and $c_k$ is of course increasing, it holds that $\mathcal{M}_2$ is valid.

As for the complexity, we can inherit the results of Proposition 2.1 directly as the main computational time lies in the subroutine Algorithm 1. So, the computational complexity is polynomial and it finishes the proof.

### C.6 Omitted Proof in Section 2.4

### C.6.1 Proof of Lemma 2.3

In order to prove that $\int_c^b y(t) \cdot dt$ is independent of the integration path for any $c$, we first write down how we decide $y(\cdot)$. From the implementation of Algorithm 1, we know that

$$y(c) = \underset{y}{\text{argmin}}\ c \cdot y$$

$$\text{s.t.}\ y_k^{1/\rho_k} + \sum_{i=1}^{K} \mu_{ik} y_i^{\gamma_{ik}/\rho_i} \geq d_k$$

$$y_k \geq 0 \text{ for any } k \in [K].$$

From the knowledge of analysis (Griffiths & Schroeter, 2019), we know that a sufficient condition is $\nabla \times y(c) = 0$. On the other hand, if we can find a potential function $\Psi(c)$ such that $\nabla \Psi(c) = y(c)$, it holds that $\nabla \times y(c) = \nabla \times \nabla \Psi(c) = 0$ due to $\nabla \times \nabla = 0$.

We use $\mathcal{I}$ to denote the constrained area that $\mathcal{I} = \{y : y_k^{1/\rho_k} + \sum_{i=1}^{K} \mu_{ik} y_i^{\gamma_{ik}/\rho_i} \geq d_k \text{ and } y_k \geq 0 \text{ for any } k \in [K]\}$. From Lemma C.3, we know that $\mathcal{I}$ is a convex set. Note that $c \cdot y$ is a linear function, then we can define $\Psi(c) = \min_{y \in \mathcal{I}} c \cdot y$. Let's now detail when we change $c$ to $c + \delta_c$, how will $y$ change. There are two possible outcomes. If $y(c)$ is a non-smooth point, then $y(c + \delta_c) = y(c)$ (ref. Figure 5). Otherwise, $y(c)$ is on a smooth segment, and we know that $c \cdot \nabla y(c) \cdot \delta_c = 0$ (ref. Figure 6). Since $\delta_c$ can be arbitrary, it holds that $c \cdot \nabla y(c) = 0$. Therefore, we know that $\Psi(c + \delta_c) - \Psi(c) = \delta_c \cdot y(c)$ in the first case and $\Psi(c + \delta_c) - \Psi(c) \to \delta_c \cdot y(c) + c \cdot \nabla y(c) \cdot \delta_c = \delta_c \cdot y(c)$ as $\delta_c \to 0$ in the second case. We know thereof that $\nabla \Psi(c) = y(c)$, showing the path independence. Since we have proven the path independence, it holds that the payment rule $p(\cdot)$ is well-defined which ends the proof.

### C.6.2 Proof of Theorem 2.3

From the revelation principle (Roughgarden, 2010), we know the existence of mechanisms satisfying both IC and IR. Let's analyze the IC condition first. It holds that $p(c) - p(\widehat{c}) \geq c \cdot y(c) - c \cdot y(\widehat{c})$. Since $\widehat{c}$ can approach $c$ from any direction, we know that $\nabla p(c) = c \cdot \nabla y(c)$. In practice, applicable mechanisms are

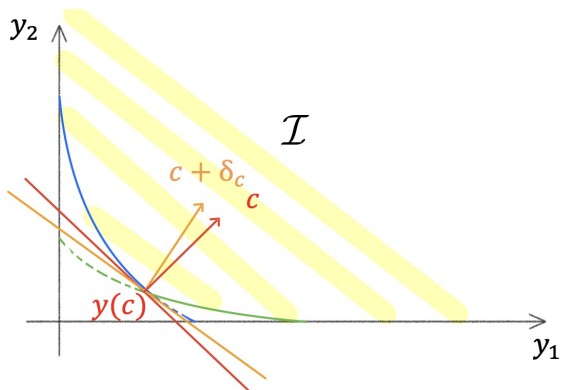

Figure 5: When $y(c)$ is a vertex.

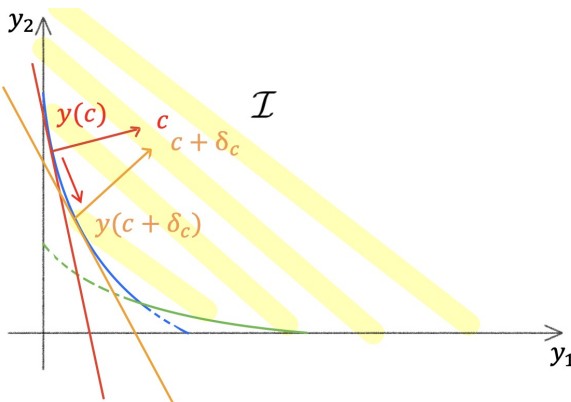

Figure 6: When $y(c)$ is not a vertex.

usually smooth, so we assume that these gradients exist. However, as we will see below the payment rule is indeed a constant, this assumption is not mandatory.

Consequently, it holds that

$$p(c) = p(b) - \int_c^b t \cdot \nabla y(t) \cdot dt = c \cdot y(c) + \int_c^b y(t) \cdot dt + C,$$

where the second equation holds due to the integration by parts formula and $C$ here is a constant.

We now know we can use this kind of payment rule to induce any allocation rule, which is thereof applied to social welfare maximizing platforms. Since it contains a family of mechanisms, we use the IR condition to find the one with the highest revenue. Recall that we need to guarantee $p(c) - c \cdot y(c) \geq 0$ for any $c$. Since there are constraints that $y_k(\cdot) \geq 0$ for any $k \in [K]$, it holds that $C \geq \max_c[-\int_c^b y(t) \cdot dt]$. When $c = b$, we know that the right-hand side takes the maximum zero. Therefore, we know the smallest $C$ is zero which is optimal.

The last question is that we only show a necessary condition and we need to prove the IC condition actually holds. From the proof of Lemma 2.3, we know that $c \cdot \nabla y(c) = 0$. Therefore, we know that $p(\cdot)$ is in fact a constant, say $b \cdot y(b)$. Then, we only need to show that $c \cdot y(c) \leq c \cdot y(\widehat{c})$. Note that from Algorithm 2, we find $y(c)$ to minimize $c \cdot y$. Thus, it holds that $c \cdot y(c) = \text{argmin}_{y \in \mathcal{I}} c \cdot y \leq c \cdot y(\widehat{c})$ as $y(\widehat{c})$ is also a feasible candidate.

Therefore, we show the payment rule in Algorithm 2 satisfies both IC and IR. Since the objective function is $\sum_k \widehat{c}_k y_k = \sum_k c_k y_k = c \cdot y$ when $c = \widehat{c}$ and $y_k = x_k^{\rho_k}$, we know that the mechanism $\mathcal{M}_4$ achieves the highest social welfare. Moreover, since $\mathcal{M}_4$ has the same allocation rule as $\mathcal{M}_2$, we know it's valid immediately.

Note that most of the computational resource consumption comes from calls to Algorithm 1, hence we know Algorithm 2 is a polynomial algorithm, which ends our proof.

We finally give the following remark. Observe that from the proof of Theorem 2.3, we don't need Assumption 2.1 anymore unlike Theorems 2.1 and 2.2. It may suggest the potential success of high-dimensional mechanism design problems beyond cycle monotonicity (Lavi & Swamy, 2007), which is of independent interest for future work.

### C.6.3 Discussion on Mechanism $\mathcal{M}_3$

The difficulty of Mechanism $\mathcal{M}_3$ stands in sharp contrast to $\mathcal{M}_4$, which maximizes social welfare. Under welfare maximization, one can leverage a potential-function-based formulation $\Psi(c)$ to construct path-independent payments. However, this approach does not apply to Mechanism $\mathcal{M}_3$. Specifically, for $\mathcal{M}_3$ we require a path-independent allocation rule $y$ that satisfies $y(c) = \arg\min c \cdot y + \int_c^b t \cdot \nabla y(t) \cdot dt$, which is

fundamentally challenging because the linear objective and the integral term are tightly coupled. Moreover, enforcing path independence, without access to a potential function (The existence of a potential function implies that the field is conservative, which naturally leads to path-independent payments), makes identifying such an allocation especially difficult. Specifically, there is no clear evidence showing that naively derived $y(c) = \arg\min c \cdot y + \int_c^b t \cdot \nabla y(t) \cdot dt$ naturally satisfies the integral path-independence property.

This comparison highlights an important advantage of social-welfare maximization: it enables simple and practically implementable mechanisms. Meanwhile, our experiments show that the welfare-maximizing mechanism only incurs a minor revenue loss, further supporting its practical relevance.

### C.7 Omitted Proof in Section 2.5

#### C.7.1 Proof of Theorem 2.4

First, from the implementation of Algorithms 2 and 4, we know that they achieve social welfare optimum as the demand constraints are unavoidable. Therefore, we know that $SW_2 = SW_4$ and they are larger than both $SW_1$ and $SW_3$, yielding $SW_2 = SW_4 \geq \max\{SW_1, SW_3\}$.

In the meanwhile, since $\mathcal{M}_1$ and $\mathcal{M}_2$ have the same constraints, then the two solutions are both feasible for each optimization problem. Thus, since $\mathcal{M}_1$ maximizes the revenue of the platform, we know that $Rev_1 \geq Rev_2$.

Now, let's compare $Rev_2$ and $Rev_4$. Since they have the same allocation rule, we only need to compare $\sum_k \int_{c_k}^{b_k} y_k(c_{-k}, t_k) dt_k$ and $\int_c^b y(t) \cdot dt$. Since we know the latter integral doesn't depend on the integration path, we choose a path as follows. We first go from $c$ to $(b_1, c_2, ..., c_K)$ and then from $(b_1, c_2, ..., c_K)$ to $(b_1, b_2, c_3, ..., c_K)$ etc. The $k$-th road is from $(b_1, ..., b_{k-1}, c_k, ..., c_K)$ to $(b_1, ..., b_k, c_{k+1}, ..., c_K)$. Therefore, it holds that $\int_c^b y(t) \cdot dt = \sum_k \int_{c_k}^{b_k} y_k(b_1, ..., b_{k-1}, t_k, c_{k+1}, ..., c_K) dt_k$. From the implementation of Algorithms 2 and 4, we know that $y_k(b_1, ..., b_{k-1}, t_k, c_{k+1}, ..., c_K) \geq y_k(c_1, ..., c_{k-1}, t_k, c_{k+1}, ..., c_K)$ as the objective function is linear and the feasible region is a convex set, yielding $\sum_k \int_{c_k}^{b_k} y_k(c_{-k}, t_k) dt_k \leq \int_c^b y(t) \cdot dt$. Thus, we know that $\mathbb{E}_c[\sum_k p_k(c)]$ in $\mathcal{M}_2$ is no larger than $\mathbb{E}_c[p(c)]$ in $\mathcal{M}_4$ and $Rev_2 \geq Rev_4$.

Note that $\mathcal{M}_3$ and $\mathcal{M}_4$ have the same constraints though different optimization objectives. From the above proof, we know that if the allocation of $\mathcal{M}_3$ is $x(\cdot)$, it holds that $p(c) \geq c \cdot y(c) + \int_c^b y(t) \cdot dt$ where $y_k(\cdot) = x_k(\cdot)^{\rho_k}$. Here, we also need that $\int_c^b y(t) \cdot dt$ is independent of the integration path.

Consequently, as $\mathcal{M}_3$ has one more constraint on the integration path than $\mathcal{M}_1$, the allocation of $\mathcal{M}_3$ is also feasible for the optimization Problem 1. Let's assume the corresponding allocation rule is associated with $y(\cdot)$.

Notice that the payment rule of $\mathcal{M}_3$ holds $p(c) \geq c \cdot y(c) + \int_c^b y(t) \cdot dt$, we only need to show that $c \cdot y + \int_c^b y(t) \cdot dt \geq c \cdot y + \sum_k \int_{c_k}^{b_k} y_k(t_k, c_{-k}) dt_k$. Since $\mathcal{M}_3$ is a valid mechanism and has path independence, we choose the following path which goes from the first coordinate to the last one in order. More specifically, the $k$-th road is from $(b_1, ..., b_{k-1}, c_k, c_{k+1}, ..., c_K)$ to $(b_1, ..., b_{k-1}, b_k, c_{k+1}, ..., c_K)$ along the $k$-th coordinate. Due to Definition 2.1, it holds that $y_k(b_1, ..., b_{k-1}, t_k, c_{k+1}, ..., c_K) \geq y_k(t_k, c_{-k})$, and then we know $\int_c^b y(t) \cdot dt \geq \sum_k \int_{c_k}^{b_k} y_k(t_k, c_{-k}) dt_k$ similar to the proof of Theorem 2.4, yielding $p(c) \geq c \cdot y + \sum_k \int_{c_k}^{b_k} y_k(t_k, c_{-k}) dt_k$.

Since the payment of $\mathcal{M}_1$ is the minimum of $c \cdot y + \sum_k \int_{c_k}^{b_k} y_k(t_k, c_{-k}) dt_k$ over all possible allocation rule $x(\cdot)$ or equivalently $y(\cdot)$, it holds that $p(c)$ is no smaller than the payment of $\mathcal{M}_1$. Therefore, the expected payment of $\mathcal{M}_3$ is at least as large as the one of $\mathcal{M}_1$. Since the platform's revenue is equal to the negative of the payment, it holds that $Rev_3 \leq Rev_1$, which ends our proof.

# D Omitted Details in Section 3

## D.1 Timeline of the Three-Layer Platform-Broker-Creator Markets Markets

We use the following figure to show the timeline in three-layer markets.

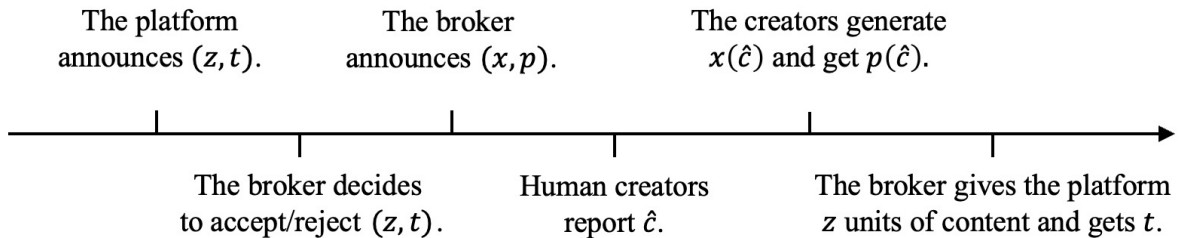

Figure 7: The timeline of the three-layer market.

## D.2 Omitted Proof in Section 3.1

### D.2.1 Proof of Theorem 3.1

First, as we notice that the final allocation in both stages is constant, we know that $\mathcal{M}_5$ is valid.

We now begin to consider the second-stage mechanism design. Note that in this stage, $t$ and $z$ are fixed constants. Therefore, changing the objective from $\mathbb{E}_c[t - \sum_k p_k(c; z, t)]$ to $\mathbb{E}_c[-\sum_k p_k(c; z, t)]$ won't change the optimal allocation and payment rule. Comparing Problem 4 and Problem 1, we know that the only change is the demand constraint is now $x_k(c; z, t) \geq z_k$, parallel to other axes. Since we know $z_k \geq 0$, we can certainly omit $x_k \geq 0$. From the proof of Theorem 2.1, we know that this optimization problem equaling to minimizing $\min_y \sum_k (c_k + \frac{F_k(c_k)}{f_k(c_k)}) y_k$ subject $y_k^{1/\rho_k} \geq z_k$ where $y_k = x_k^{\rho_k}$. Since $c_k + \frac{F_k(c_k)}{f_k(c_k)} \geq 0$, it holds that the optimal $y_k(c; z, t)$ is $z_k^{\rho_k}$, hence $x_k(c; z, t) = z_k$ for any $c$. Recall that due to Assumption 2.3, $c_k + \frac{F_k(c_k)}{f_k(c_k)}$ is increasing, so we have validity directly. In the meanwhile, we know that $p_k(c; z, t) = c_k y_k(c; z, t) + \int_{c_k}^{b_k} y_k(\tau_k, c_{-k}; z, t) d\tau_k$, using the fact that $y_k(c; z, t) = z_k^{\rho_k}$ for all $c$, we know that $p_k(c; z, t) = b_k y_k(c; z, t) = b_k z_k^{\rho_k}$. Since $b_k \geq c_k$, the IR constraint is preserved.

The third step is to find the optimal first-stage mechanism. Since we need to satisfy $\mathbb{E}_c[t - \sum_k p_k^*(c; z, t)] \geq 0$ and from above we know that $p_k^*(c; z, t) = b_k y_k(c; z, t) = b_k z_k^{\rho_k}$. So, we know that $t \geq \sum_k b_k z_k^{\rho_k}$. As the objective is to maximize $\mathbb{E}_c[-t]$, it equals to minimizing $\sum_{k=1}^K b_k \omega_k$ where we use $\omega_k$ to denote $z_k^{\rho_k}$. Now we turn to the constraints. It holds that $\omega_k^{1/\rho_k} + \sum_{i=1}^K \mu_{ik} \omega_i^{\gamma_{ik}/\rho_i} \geq d_k$ and $\omega_k \geq 0$ for any $k \in [K]$. From the implementation of Algorithm 1, we know that we only need to replace $vc_k(\hat{c})$ by $b_k$, and $\omega$ is the output $y$ of Algorithm 1.

Therefore, we prove the optimality of the second-stage mechanism design using $x_k(c; z, t) = z_k$ and $p_k(c; z, t) = b_k z_k^{\rho_k}$. Also, we know that the allocation rule in $\mathcal{M}_5$ is optimal. Since $t$ can be any number no smaller than $\sum_{k=1}^K b_k z_k^{\rho_k}$, we know that the payment rule in the first stage is optimal as well.

Finally, since the time complexity mainly depends on the call of Algorithm 1, we know $\mathcal{M}_5$ is polynomial-time from Proposition 2.1 immediately.

## D.3 Omitted Proof in Section 3.2

### D.3.1 Proof of Theorem 3.2

Note that the second-stage optimization problem is the same as the one of Theorem 3.1. We only need to replace the first-stage problem with Problem 6. Now, the objective is to minimize $\mathbb{E}_c[-\sum_k c_k(x_k^*(c; z, t))^{\rho_k}]$.

Since we know that $x_k^*(c; z, t) = z_k$, it holds that we need to minimize $\mathbb{E}[\sum_k c_k z_k^{\rho_k}]$. Using $\omega_k$ to denote $z_k^{\rho_k}$, it holds that the objective is now $\mathbb{E}[c] \cdot \omega$ under the constraints $\omega_k^{1/\rho_k} + \sum_{i=1}^{K} \mu_{ik} \omega_i^{\gamma_{ik}/\rho_i} \geq d_k$ and $\omega_k \geq 0$ for any $k \in [K]$. Replacing $vc_k(\widehat{c})$ by $\mathbb{E}[c_k]$ yields the optimal $\omega$ immediately.

Since we need to guarantee $t \geq \sum_k b_k z_k^{\rho_k}$ so that the broker will participate in the market, we set $t = \sum_{k=1}^{K} b \cdot \omega$, which is the lower bound of feasible $t$. Therefore, it holds that $(z, t)$ in $\mathcal{M}_6$ is the optimal first-stage mechanism.

Similarly, we know that the corresponding $(x, p)$ is the optimal second-stage mechanism. Since the allocation is fixed in each stage, we know that $\mathcal{M}_6$ is valid. As for the computational complexity, it's polynomially dominated by the one of Algorithm 1. Therefore, we know it's a polynomial-time algorithm from Proposition 2.1 which ends the proof.

### D.4 Omitted Proof in Section 3.3

### D.4.1 Proof of Theorem 3.3

From Theorem 2.4, we know that $Rev_2 \geq Rev_4$. Also, in the proof of Theorem 2.2, we know that the payment in $\mathcal{M}_4$ is in fact a constant $b \cdot y(b)$ where $y(b)$ is the output of Algorithm 1 with input $b$. From the implementation of Algorithm 3, we know that the payment of $\mathcal{M}_5$ is also $b \cdot y(b)$ thereof $Rev_4 = Rev_5$. Note that $\mathcal{M}_5$ and $\mathcal{M}_6$ have the same second-stage optimization problem, therefore $\mathcal{M}_6$ is also feasible for a Type 1 platform. Due to the optimality of $\mathcal{M}_5$ shown in Theorem 3.1, we know that $Rev_5 \geq Rev_6$. Hence, it holds that $Rev_2 \geq Rev_4 = Rev_5 \geq Rev_6$.

Similarly, as $\mathcal{M}_5$ is feasible for a Type 2 platform, we know that $SW_6 \geq SW_5$ because of Theorem 3.2. Note that $\mathcal{M}_2$ and $\mathcal{M}_4$ achieve the highest social welfare for each $c$ while $SW_6$ only uses the optimal allocation with respect to $\mathbb{E}[c]$. Therefore, for each $c$, the allocation of $\mathcal{M}_2$ and $\mathcal{M}_4$ is at least as good as the one of $\mathcal{M}_6$. We then know that $SW_2 = SW_4 \geq SW_6$. To sum up, it holds that $SW_2 = SW_4 \geq SW_6 \geq SW_5$.

As a result, we've shown that the three-layer market will lead to a lose-lose situation that

$$\max\{Rev_5, Rev_6\} \leq \min\{Rev_2, Rev_4\}$$

and

$$\max\{SW_5, SW_6\} \leq \min\{SW_2, SW_4\}.$$

### D.4.2 Proof of Corollary 3.1

We only need to construct a mechanism based on $\mathcal{M}_5$ which is feasible for Problem 3. We choose the second-stage mechanism in Algorithm 3 as a candidate. We know that the total payment of $\mathcal{M}_5$ is now $\sum_k b_k z_k^{\rho_k}$ and the allocation rule is a constant vector $z$. Then, we choose $p(c) = \sum_{k=1}^{K} b_k z_k^{\rho_k}$ and $x(c) = z$ for Problem 3.

For Problem 3, since we know $z$ also satisfies demand constraints in all domains, it holds that $x_k(c) + \sum_{i=1}^{K} \mu_{ik} x_i(c)^{\gamma_{ik}} \geq d_k$ also holds. For the IC constraint, we know that $p(c) - \sum_{k=1}^{K} c_k x_k^{\rho_k}(c) = \sum_{k=1}^{K} b_k z_k^{\rho_k} - \sum_{k=1}^{K} c_k z_k^{\rho_k} = p(\widehat{c}) - \sum_{k=1}^{K} c_k x_k^{\rho_k}(\widehat{c})$ so preserved. For the IR constraint, it holds that $p(c) - \sum_{k=1}^{K} c_k x_k^{\rho_k}(c) = \sum_{k=1}^{K} b_k z_k^{\rho_k} - \sum_{k=1}^{K} c_k z_k^{\rho_k} = \sum_{k=1}^{K} (b_k - c_k) z_k^{\rho_k} \geq 0$ as $b_k \geq c_k$. Finally, $x_k(c) \geq 0$ for all $k \in [K]$ as $z \geq 0$ entrywise.

Therefore, we find a feasible solution to Problem 3. Since $Rev_3$ corresponds to the optimal solution, we know that $Rev_3 \geq \sum_{k=1}^{K} b_k z_k^{\rho_k} = Rev_5$, yielding a lower bound for $Rev_3$, and it finishes the proof.

## E Omitted Details in Section 4

We conduct all the numerical experiments written in Python 3.11.7 running on a laptop with an Apple M2 CPU, and we provide more omitted details as follows.

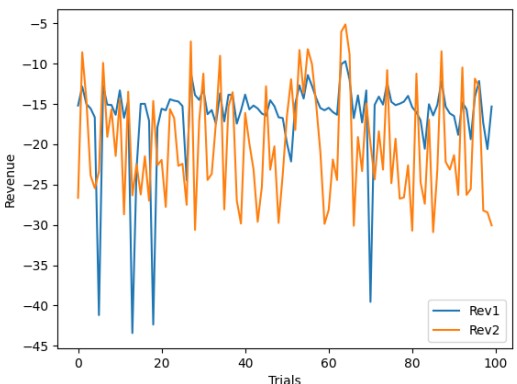 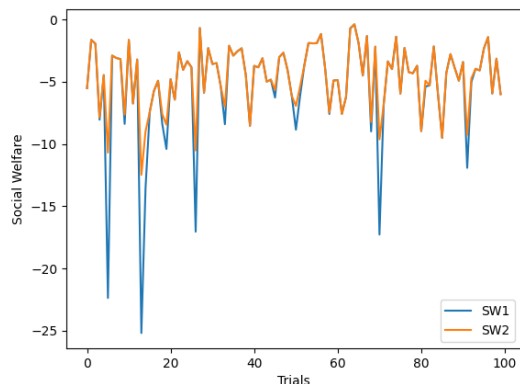

(a) Comparative platform revenue volatility for $\mathcal{M}_1$ and $\mathcal{M}_2$ over 100 trials.
(b) Comparative social welfare volatility for $\mathcal{M}_1$ and $\mathcal{M}_2$ over 100 trials.

Figure 8: Volatility comparison between Mechanism 1 and Mechanism 2.

**Experimental evaluation of revenue and welfare among mechanisms.** For series of experiments to compare the performance of Algorithms 1 to 3, 4 and 5, we randomly choose $\mu$ following Unif$[0, 0.5]$, $\gamma$ following Unif$[0.8, 0.9]$, $d$ following Unif$[2, 5]$ and $\rho$ following Unif$[1, 1.5]$. To better visualize our results, we still set $K = 2$. We introduce asymmetry that $c_1$ follows $\mathcal{N}(0, 1)$ truncated by $[0, 10]$ and $c_2$ is sampled from Unif$[0, 1]$. We follow Equation (7) and use 100 trials to take the average in order to reduce the variance of our payment rule. Finally, we sample the cost vector $c$ 1000 times and compare the corresponding revenue and social welfare. Here, we compare the resulting revenue and social welfare under different market structures and objective choices, holding the distribution of costs $c$ (i.e., market randomness) and other hyperparameters fixed across all mechanisms. Therefore, the differences observed are solely caused by structural changes (e.g., two-layer vs. three-layer, revenue-maximizing vs. welfare-maximizing), which makes the comparison meaningful.

We first give the following Table 1 to show the average revenue and social welfare of $\mathcal{M}_1$, $\mathcal{M}_2$, $\mathcal{M}_4$, $\mathcal{M}_5$ and $\mathcal{M}_6$, and we use Theorem 2.4 and Corollary 3.1 to bound the ones with $\mathcal{M}_3$.

Table 1: Revenue and social welfare of our mechanisms.

|  | $\mathcal{M}_1$ | $\mathcal{M}_2$ | $\mathcal{M}_3$ | $\mathcal{M}_4$ | $\mathcal{M}_5$ | $\mathcal{M}_6$ |
|---|---|---|---|---|---|---|
| Revenue | -15.96 | -20.28 | [-30.68,-15.96] | -30.68 | -30.68 | -34.21 |
| Social Welfare | -5.24 | -4.79 | $\leq$ -4.79 | -4.79 | -9.58 | -4.96 |

Note that the revenue is the negative payment, and the social welfare is the negative total human cost, so they are both less than zero. The experimental findings corroborate Theorems 2.4 and 3.3, demonstrating that the rankings of revenue and social welfare observed in the experiments are consistent with their theoretical rankings.

We visualize in Figure 4 the distributions of revenue and social welfare across different algorithms and provide further explanation and economic insights. We use a red line to represent the mean and an orange one for the median. Moreover, we draw the first quantile, the third quantile, and the extension with a 1.5 interquartile range in Figure 4. We find that compared with mechanism $\mathcal{M}_2$, mechanism $\mathcal{M}_1$ has more outliers and extreme points. We also plotted the specific $Rev_1$, $Rev_2$, $SW_1$ and $SW_2$ for the first 100 trials in Figure 8, which clearly demonstrated the high volatility of mechanism $\mathcal{M}_1$. It may hint that the social welfare maximizer can enjoy a more stable market, say the social welfare and even the revenue are steady, compared with a revenue maximizer. This could perhaps explain why some companies turn to maximizing social welfare. Moreover, in the three-layer market, data brokers and online platforms first establish contracts,

which are independent of the realization value of cost $c$, resulting in less volatility. This is also clearly reflected in the box plot.

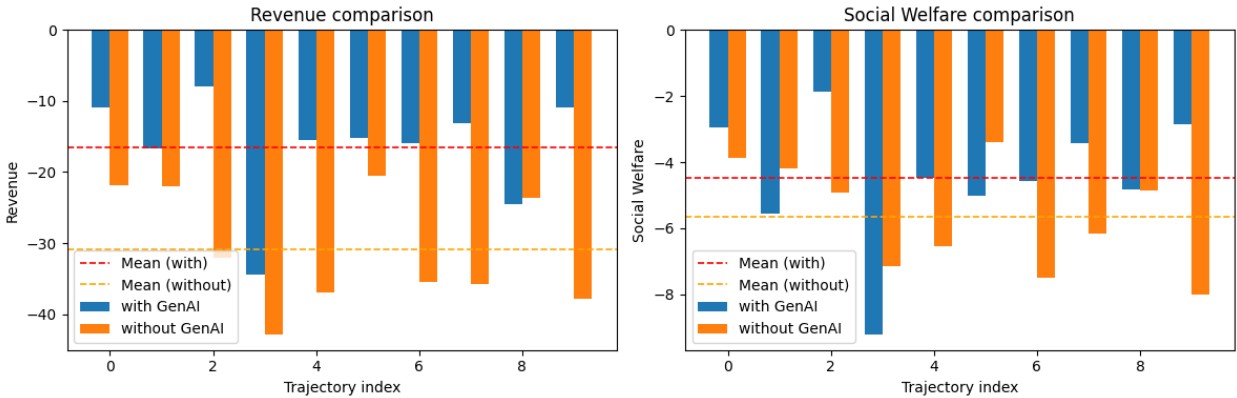

Figure 9: Visualization of *Rev* and *SW* with and without GenAI (lower values mean worse performance).

**With and Without GenAI.** Finally, we provide more details when studying the gap between the presence and absence of GenAI. Recall that when $\mu = 0$ is equivalent to expelling GenAI out of the market. We adopt the same setup as in the evaluation of the mechanisms. The experiment is repeated 10 times, and the results are reported in Table 2. Besides, we include bar plots to visualize their gaps in Figure 9.

Table 2: Revenue and social welfare with and without GenAI in 10 trajectories.

| | | | | | | | | | | | |
|---|---|---|---|---|---|---|---|---|---|---|---|
| with GenAI | $Rev$ | -10.88 | -16.65 | -7.98 | -34.49 | -15.55 | -15.24 | -15.88 | -13.18 | -24.51 | -10.87 |
| | $SW$ | -2.94 | -5.54 | -1.86 | -9.22 | -4.47 | -5.01 | -4.58 | -3.42 | -4.81 | -2.86 |
| without GenAI | $Rev$ | -21.82 | -21.95 | -32.08 | -42.91 | -36.93 | -20.57 | -35.46 | -35.83 | -23.68 | -37.80 |
| | $SW$ | -3.86 | -4.20 | -4.93 | -7.13 | -6.54 | -3.40 | -7.49 | -6.15 | -4.84 | -8.02 |

The average revenue increases from -30.90 to -16.52 while the average social welfare increases from -5.66 to -4.47. Therefore, the revenue and social welfare increase by 46.54% and 21.02%, respectively. This indicates that the application of GenAI moderately improves production efficiency, while simultaneously increasing both the platform's revenue and overall social welfare.

