# OpenReview forum: "GenAI vs. Human Creators: Procurement Mechanism Design in Two-/Three-Layer Markets"
_TMLR — Accepted by TMLR_

### Review · Reviewer_HWUu · 2025-11-25

**Summary Of Contributions:**

The paper develops optimal procurement mechanisms in markets where platforms source content from both human creators and GenAI, under nonlinear production technologies and cross-domain transferability. Its main contributions are:
1. Two-layer market mechanisms: Derivation of optimal revenue-maximizing (M1) and welfare-maximizing (M2) mechanisms, extending the classical analysis to incorporate non-linear terms arising from GenAI creators. Also, formulating the problem of designing creator-union mechanisms.
2. Three-layer market: Providing optimal mechanisms for both objectives (M5, M6), and identifying a “lose–lose” effect, where brokers reduce both platform revenue and welfare.
3. Simulation results supporting qualitative predictions.

Some noticeable weaknesses include:
1. Modeling assumptions (e.g., functional forms of GenAI transferability) may be strong or narrow.
2. Revenue-maximizing multidimensional case (M3) remains incomplete and open, as pointed out by the authors.
3. Simulations are synthetic; empirical grounding is limited.

**Audience:**

Yes

**Audience Explanation:**

The paper addresses an increasingly important topic at the intersection of mechanism design, AI economics, and content-generation markets. The paper provides new theoretical mechanisms tailored to environments involving GenAI transferability, which is a novel modeling consideration likely to interest both ML researchers and economic mechanism designers in the TMLR community.

**Claims And Evidence:**

Yes

**Claims Explanation:**

The paper’s primary theoretical claims are backed by formal derivations with stated assumptions; the mechanisms (M1~M6) follow from well-defined optimization problems, convexity arguments, and standard incentive-compatibility constructions.

**Requested Changes:**

1. Justifying the cost of GenAI is "negligible," compared to human effort. This might be true only in some domains, at least in the present.
2. More discussion around $\mathcal{M}_3$ can be provided, for instance, in Appendix. Difficulties and failed attempts are illustrative enough to provide more context on the exact challenges.

---

> ### Author Response · Authors · 2026-01-07
>
> Thank you for the encouraging feedback and constructive questions. We believe our response below should help thoroughly address the reviewer's major questions, and are happy to engage with the reviewer if there is any further question.
>
> *Re Modeling assumptions*: We believe our choices of the power-law structure and the transferability assumption are well-justified. First, these assumptions are widely adopted and recognized in the literature: for example, [1] studies the scaling laws of LLMs, and [2] similarly models how GenAI can generate new content units based on human-created content to analyze the symbiosis or conflict between GenAI and human creators. Second, this representative and minimal model enables us to derive closed-form characterizations and FPTAS-based mechanisms. They further demonstrate (see Theorems 2.1, 2.3, 3.1, and 3.2) that the market competition between GenAI and human creators is causing pressing issues. A promising direction for future work is to extend our analysis to more general production technologies for GenAI—e.g., increasing, concave, or regular contributions—and examine whether our key comparative statics, such as overproduction and the lose-lose outcome, continue to hold. Our paper is the first to identify the three-layer lose-lose phenomenon, and we intentionally adopt a simple yet representative framework to draw the community’s attention to this emerging issue and to open up new avenues for research.
>
> [1] Jared Kaplan, Sam McCandlish, Tom Henighan, Tom B Brown, Benjamin Chess, Rewon Child, Scott Gray,
> Alec Radford, Jeffrey Wu, and Dario Amodei. Scaling laws for neural language models. arXiv preprint
> arXiv:2001.08361, 2020.
>
> [2] Fan Yao, Chuanhao Li, Denis Nekipelov, Hongning Wang, and Haifeng Xu. Human vs. generative ai in
> content creation competition: Symbiosis or conflict? arXiv preprint arXiv:2402.15467, 2024.
>
> *Re The cost of GenAI is negligible*: We use the term "negligible" in a relative sense—specifically, relative to the per-unit cost of human creative labor, especially in scenarios of marginal content reuse. For example, once a large model such as Sora 2 is already trained, producing a new video takes only a few minutes of computation, whereas professional human creators require substantial compensation and effort to produce comparable content.
>
> Moreover, we would like to emphasize that our model naturally extends to settings with small but non-zero GenAI costs. We assume the training cost of GenAI is proportional to human production cost, and the training cost parameter of GenAI is $c^G_k$ in domain $k$. Consider Mechanism 1 as an example: our transformation is derived entirely from IC and IR constraints, independent of the objective function. Therefore, incorporating GenAI costs is equivalent to adding $\sum_{k=1}^K c^G_k y_k$ to the platform’s objective. As a result, we only need to adjust the virtual cost from $vc_k$ to $vc_k+c^G_k$ to recover the optimal mechanism under non-zero GenAI costs. The same modification applies to other mechanisms as well. For instance, in Mechanism 2, replacing $\hat c_k$ with $\hat c_k+c_k^G$ leaves the structure and insights unchanged. We include this extension in the Appendix C.1 for completeness.

---

> > ### Author Response · Authors · 2026-01-07
> >
> > *Re More discussion around $\mathcal{M}_3$*: First, this is a well-recognized difficulty in the literature: revenue-maximizing mechanism design in a multidimensional environment is an open problem [3,4,5], and, to date, no closed-form characterization is known beyond a few very special cases. In contrast, we explicitly formulate the optimization problem for Mechanism 3 and provide both revenue and social-welfare upper and lower bounds in Theorem 2.4.
> >
> > Second, the difficulty of Mechanism 3 stands in sharp contrast to Mechanism 4, which maximizes social welfare. Under welfare maximization, one can leverage a potential-function–based formulation to construct path-independent payments. However, this approach does not apply to Mechanism 3. Specifically, for $\mathcal{M}_3$ we require a path-independent allocation rule $y$ that satisfies $y(c)=\arg\min c\cdot y+\int_c^b t\cdot \nabla y(t)\cdot dt$, which is fundamentally challenging because the linear objective and the integral term are tightly coupled. Moreover, enforcing path independence — without access to a potential function — makes identifying such an allocation especially difficult.
> >
> > This comparison highlights an important advantage of social-welfare maximization: it enables simple and practically implementable mechanisms. Meanwhile, our experiments show that the welfare-maximizing mechanism only incurs a minor revenue loss, further supporting its practical relevance. We add more discussion on $\mathcal{M}_3$ in Appendix C.6.3.
> >
> > [3] Patrick Briest, Shuchi Chawla, Robert Kleinberg, and S Matthew Weinberg. Pricing randomized allocations.
> > In Proceedings of the twenty-first annual ACM-SIAM symposium on Discrete Algorithms, pp. 585–597.
> > SIAM, 2010.
> >
> > [4] Constantinos Daskalakis. Multi-item auctions defying intuition? ACM SIGecom Exchanges, 14(1):41–75,
> > 2015.
> >
> > [5] Sergiu Hart and Noam Nisan. Approximate revenue maximization with multiple items. Journal of Economic
> > Theory, 172:313–347, 2017.
> >
> > *Re Simulations are synthetic*: Thank you for your comment. This work is primarily theoretical, and the simulations are intended to illustrate the implications of our mechanisms rather than to serve as predictive evaluation. The primary rationale for this choice is that, empirical validation is currently challenged by data availability: real GenAI procurement contracts and cross-domain training data are largely proprietary. Nevertheless, our simulations systematically sweep the parameter space (multiple combinations of $(c,\mu,\gamma,\rho,d)$), demonstrating that the key comparative statics established in our theorems — such as overproduction and the lose-lose outcome — are robust across a wide range of settings. These results highlight important regulatory implications. In future work, we aim to extend our analysis to real-world data as access becomes feasible.

---

### Review · Reviewer_k4Hf · 2025-11-29

**Summary Of Contributions:**

This paper studies nonlinear procurement mechanism design in markets where a platform acquires content from human creators and GenAI, introducing cross-domain transferability of data. The authors analyze six mechanism variants (M1–M6) across different market structures. A core contribution is to capture how content in one domain enables GenAI to generate content in another domain.

**Additional Comments:**

Although, I can see that M1 is convex and has a single optimum w.r.t. $y$, it is still non-linear w.r.t. $x$ and can exhibit multiple solutions. Is that correct (If I am mistaken please correct me)? If so, how does that affect the analysis?

**Audience:**

No

**Audience Explanation:**

TMLR’s readership is unlikely to find this paper aligned with the venue’s core interests. TMLR focuses on empirical machine learning, theoretical work grounded in learning algorithms, and analyses of artificial or biological learning systems

(https://jmlr.org/tmlr/editorial-policies.html#:~:text=Transactions%20on%20Machine%20Learning%20Research,artificial%20or%20biological%20learning%20systems;),

whereas this submission is fundamentally an economics/game-theoretic paper centered on pricing digital goods, platform revenue optimization, multidimensional incentive-compatibility constraints, and transferability as an economic property. Although the authors motivate their work through GenAI, the actual contributions lie in mechanism design and economic modeling rather than in machine learning theory: there is no learning algorithm, no optimization method beyond a convex reduction, no ML benchmarking, no generalization insight, no modeling contribution to GenAI systems, and no theoretical advance in ML. The connection to machine learning is therefore minimal and largely thematic. For these reasons, this work may not fit TMLR’s intended audience and would be more appropriate for venues in computational economics, econometrics, operations research etc.

**Broader Impact Concerns:**

I do not see any ethical implications that would require adding a Broader Impact Statement.

**Claims And Evidence:**

No

**Claims Explanation:**

The claims are partially convincing to me. For example, while I am convinced on items such as:

- I can see that there can be information flow from different domains that boost the productivity of AI systems.

I am not sure that I am convinced:

- That the transferability is absent in other goods. For example, regarding "R&D spillovers": Innovation in one sector increases productivity in another (e.g., semiconductor advances benefit healthcare imaging. Another e.g., is the skills learned when a company builds a car engine is transferable to building jet engines --- a famous instance is Rolls-Royce). In a broader sense, once can also say technology transfers (in between countries), when considered as a "good", do have similar effects.

- I am also not completely convinced by the argument $\rho < 1$: This is because one can also argue that as generative AI continues to advance, the effective cost of producing novel ideas decreases. In my own experience, I can now explore many directions in parallel and quickly identify the most promising ones—something that would have required substantially more time before.

- Scaling laws look like a plausible assumption, but not validated in the context of cross-domain content production.

**Requested Changes:**

If this paper were to be published in TMLR, I recommend:

**(a) Add a detailed background / primer**

Many readers will not be familiar with:

- virtual costs,
- reverse hazard rates,
- incentive compatibility (IC)  (does that simply mean "the utility the creator gets when reporting truthfully must be at least as high as the utility they get by not. In the notation, diverging from truth is denoted as $\hat{c}_k$, if so, such an elaboration could be useful)
- path independence,
- Myerson’s analysis: I am not sure if the readers of TMLR will be familiar with this.
- convex procurement formulations,
- “valid mechanisms” monotonicity conditions.

These concepts may be standard in economic theory but not in ML. Since the audience of TMLR is ML focus, it would be nice to have a background section where you clarify these.

**(b) Clarify economic meaning of parameters**

- $x_k$ is defined as the amount of content created in domain k. Then, what does $x_k^\rho$ for $\rho < 1$? Is $x_k$ assigned to creators by the platforms?
- What are $\tilde{c}$?

**(c) Figures and results**

- Figure 4 is particularly unclear to me. I am not sure what the red and orange lines stand for. If the punchline is "both revenue and social welfare increase in the presence of GenAI.", which I see at Table 2, then it would be useful to see plots that support this claim (perhaps present the difference, rather than the objectives)

- I am note sure what "comparing the **performance** of Algorithms 1 to 3, 4 and 5" means in this setting. These are different models/assumptions that can't be verified. In other words, as far as I understand, a ground truth (held-out) data does not exists that would allow you compare the performance of these approaches. So I am not sure what "performance comparison" means in this setting.

---

> ### Author Response · Authors · 2026-01-07
>
> Thank you for the insightful remarks and questions. We believe our response below should help thoroughly address the reviewer's major questions, and are happy to engage with the reviewer if there is any further question.
>
> *Re Transferability in other goods*: This is an insigtful comment, but the example of semiconductor innovation benefiting healthcare imaging does not reflect the type of transferability we model: the upstream and downstream physical goods are still produced separately and traded in a standard vertical relationship. However, in our setting, **platforms purchase human-generated data to directly satisfy demand and simultaneously serve as productive input for GenAI**. In the literature, cross-domain effects (e.g., R&D spillovers) are typically treated as externalities outside allocation constraints, so mechanism design remains essentially low-dimensional. Our contribution is to embed transferability parameters $\mu$ and $\gamma$ into the production constraints, creating a multi-domain coupled mechanism design problem that is substantially more challenging. This structure yields: (i) dimensionality reduction enabling closed-form optimal social welfare solutions (but not for revenue maximizers in Mechanism 3), and (ii) new phenomena such as over-production, and lose-lose outcomes. Similarly, the Rolls-Royce case concerns transferability across two products, but remains different from our setting: human creations here simultaneously (i) satisfy consumer demand within their own domain, (ii) improve GenAI performance **in the same domain**, and (iii) transfer to benefit GenAI in other domains with negligible cost. This three-way role of the same content is precisely the stronger, systematic cross-domain transferability we study.
>
> In summary, our setting captures that each unit of content in domain $k$ immediately changes the platform’s production constraints across all domains, and the procurement mechanism **endogenously** exploits this structure. We study a structured ML-motivated form of transferability that is explicitly embedded in the mechanism design problem. By contrast, traditional R&D spillovers or skill transfer are typically modeled as macro or long-run externalities (e.g., productivity improvements or cost reductions), and therefore their procurement models do not appear in such form. Moreover, we focus on GenAI’s low marginal cost and its multi-area transferability across inherently distinct domains (text, images, video, code, etc). In standard procurement models for physical goods, cross-domain transfer effects are secondary and exogenous—usually abstracted as productivity shocks or externalities—rather than, as in our model, becoming a first-order element of the design constraints.
>
> *Re Misunderstanding of $\rho$'s domain*: We believe there might be a misunderstanding: throughout the paper we assumed $\rho>1$ (see Section 2.1). This captures that, within the same topic, generating multiple ideas becomes progressively slower—e.g., the first idea arrives quickly, but producing the second takes longer, which is the standard diminishing marginal productivity assumption widely used in the literature. Thus, $\rho>1$ does not compare productivity "with vs. without GenAI", but instead describes how, conditional on a given environment, idea generation within a domain exhibits decreasing marginal returns. This aligns with the classic formulations in [1,2].
>
> [1] Jagadeesan, Meena, Nikhil Garg, and Jacob Steinhardt. "Supply-side equilibria in recommender systems." Advances in Neural Information Processing Systems 36 (2023): 14597-14608.
>
> [2] Fan Yao, Chuanhao Li, Denis Nekipelov, Hongning Wang, and Haifeng Xu. "Human vs. Generative AI in Content Creation Competition: Symbiosis or Conflict?." International Conference on Machine Learning. PMLR, 2024.
>
> *Re Cross-domain scaling laws*: With the rapid development of LLMs/GenAI, there are already empirical papers supporting this assumption. For example, [3] shows that adding code data improves mathematical reasoning, with empirical evidence on the GSM8K dataset; they also define how data from domain $k$ yields greater-than-expected scaling — i.e., a synergistic effect with benchmark $j$. This cross-domain effect plays a similar role as our transferability parameter $\gamma$.
>
> [3] Hamidieh, Kimia, Lester Mackey, and David Alvarez-Melis. "Domain-aware scaling laws uncover data synergy." NeurIPS 2025 Workshop on Evaluating the Evolving LLM Lifecycle: Benchmarks, Emergent Abilities, and Scaling. 2025.

---

> > ### Author Response · Authors · 2026-01-07
> >
> > *Re TMLR interests*: We respectfully disagree with the comment that our manuscript does not fit the TMLR’s scope well. TMLR considers **experimental and/or theoretical studies yielding new insight into the design and behavior of learning in intelligent systems**, as well as **surveys that draw new connections, highlight trends, and suggest new problems in an area**. Our paper examines a GenAI-enabled multi-domain content platform as an artificial learning ecosystem involving training data, inference, cross-domain transferability, and scaling-law behavior. The core question we study — how a platform should price and procure data across domains to shape the GenAI’s capability distribution while balancing revenue and creator incentives — is fundamentally a training-data acquisition and deployment design problem. As GenAI reshapes market interactions, understanding its impact on human creators is crucial. Our results further reveal lose-lose outcomes that motivate platform design and policy considerations, offering insights we believe are directly relevant to the ML/AI community.
> >
> > Moreover, TMLR has **already** published work combining incentive design with learning systems. For exqmple, [4]  studies how to design incentives so that a learning system performs better in the presence of strategic agents and [7] studies how to design reward incentivizes in Markov Decision Processes (MDP). Similar to [4,7], we conduct economic modeling and incentive design as well, albeit for different machine learning setup, i.e., GenAI-based content generation systems in which individual agents are strategic data providers (i.e., content creators). These works all inherently **bring economic toolkit to tackle ML challenges**. Likewise, ML venues such as ICML have also published related papers (e.g., [2]) studying symbiosis/competition between GenAI and humans from another angle. Even the editorial board includes AEs whose research interests explicitly list “incentive mechanism design”, indicating that this intersection is welcome.
> >
> > In summary, we believe several segments of the TMLR readership would find our paper relevant: (i) ML researchers studying GenAI ecosystems and governance, especially training-data economics — there is already work such as [5] that explicitly positions "data economics" as part of the ML research agenda. (ii) Researchers working on mechanism design for LLMs / data markets / strategic ML. An example is [6] published at The Web Conference. (iii) Researchers in multi-agent RL, incentive design, and resource allocation.
> >
> > [4] Chen, Yatong, Jialu Wang, and Yang Liu. "Learning to incentivize improvements from strategic agents." Transactions on Machine Learning Research (2023).
> >
> > [5] Oderinwale, Hamidah, and Anna Kazlauskas. "The Economics of AI Training Data: A Research Agenda." arXiv preprint arXiv:2510.24990 (2025).
> >
> > [6] Duetting, Paul, et al. "Mechanism design for large language models." Proceedings of the ACM Web Conference 2024. 2024.
> >
> > [7] Ma, Haoxiang, et al. "Adaptive Incentive Design for Markov Decision Processes with Unknown Rewards." Transactions on Machine Learning Research (2025).

---

> > > ### Author Response · Authors · 2026-01-07
> > >
> > > *Re Background / primer*: We appreciate this constructive suggestion. In response, we have added a short section in Appendix A.2 to bridge ML and Econ readers.
> > >
> > > 1. Virtual costs: A variant of the original cost, formally defined as $c+\frac{F}{f}$. Because the platform must incentivize creators to report their true costs, the effective cost the platform faces is the human creator’s production cost $c$ plus the incentive-related virtual cost $\frac{F}{f}$.
> > > 2. Reverse hazard rates: Originating from survival analysis, and defined in this paper as $\frac{f}{F}$. It is used to characterize the log-concavity of $F$.
> > > 3. Incentive compatibility: The terminology for a desirable mechanism property that for any creator with private cost $c$, truthfully reporting $c$ to the platform always maximizes the creator’s expected utility (payment minus cost). Hence incentive compatible mechanisms can help avoid dishonest agent behaviors during information elicitation.
> > > 4. Path independence: When payments are defined through a path integral (e.g., $\int_{\hat c}^b y(t)\cdot dt$ in Mechanism 4), we require that the integral yields the same value for any path from $\hat c$ to $b$.
> > > 5. Myerson’s analysis: This refers to a classic and well-known analysis framework of [8] that converts payments into integrals of the allocation functions, which reduces mechanism design to allocation rule design.
> > > 6. Convex procurement: We hope the feasible set induced by demand constraints to be convex, so that the optimal mechanism can be computed efficiently in practice.
> > > 7. Valid mechanisms: A creator’s higher cost should lead the platform to procure less from them, while other creators’ higher costs should lead the platform to procure more from this creator. This reflects substitutability in production.
> > >
> > > [8] Myerson, Roger B. "Optimal auction design." Mathematics of operations research 6.1 (1981): 58-73.
> > >
> > > *Re Economic meaning of parameters*: $x_k$ is the amount of content the platform procures from creators in domain $k$. In Definition 2.1, $\tilde c$ denotes an arbitrary misreported cost. Incentive compatibility requires that for a creator with true cost $c_k$, any report $\tilde c_k$ cannot yield a higher utility than truthfully reporting $c_k$. Therefore, such a mechanism ensures that creators have no incentives to misreport private information.
> > >
> > > *Re Figures and results*: Thank you for the comment. While the descriptions of the red (mean) and orange (median) lines were included in the appendix, we have now also clarified them directly in the figure caption in the main text. In addition, we added bar plots (Figure 9) for revenue and social welfare to clearly show the improvements brought by GenAI.
> > >
> > > "Performance comparison" here refers to comparing the resulting revenue and social welfare under different market structures and objective choices, holding the distribution of costs $c$ (i.e., market randomness) fixed across all mechanisms. Therefore, the differences observed are solely caused by structural changes (e.g., two-layer vs. three-layer, revenue-maximizing vs. welfare-maximizing), which makes the comparison meaningful. For example, Figure 4 shows that **switching from a two-layer revenue-maximizing mechanism to a three-layer one (Mechanism 1 vs. Mechanism 5), while keeping all market conditions unchanged**, leads to decreases in both revenue and social welfare. We add more discussion in Appendix E.
> > >
> > > *Re Unique optimal $x$*: We clarify that the statement about existence of multiple optimal solutions is not correct. Since the relationship between $y$ and $x$ is monotonic and one-to-one (specifically $x_k=y_k^{1/\rho_k}$), the single optimum in $y$ directly implies a single optimum in $x$. Therefore, nonlinearity in $x$ does not introduce multiple solutions, and the analysis is unaffected.

---

### Review · Reviewer_ELEV · 2026-01-02

**Summary Of Contributions:**

This paper studies nonlinear procurement mechanism design in content markets where online platforms employ both human creators and generative AI (GenAI). The key novelty is modeling data transferability, the idea that content from one domain can improve GenAI performance in other domains. The authors analyze two market structures: a two-layer (platform-creator) market and a three-layer (platform-broker-creator) market, and derive optimal incentive-compatible and individually rational mechanisms for both revenue-maximizing and welfare-maximizing platforms. The main findings include: (1) human creators remain indispensable even with advanced GenAI, (2) overproduction occurs in domains with high data transferability, and (3) for three-layer markets, the introduction of intermediaries can distort incentives, reducing both platform revenue and social welfare. Numerical simulations validate qualitative insights.

**Audience:**

Yes

**Audience Explanation:**

This is an interesting paper with strengths shown above.

**Claims And Evidence:**

Yes

**Claims Explanation:**

**Strengths:**

S1. Novel and timely problem formulation.  The paper tackles the economic architecture of the GenAI ecosystem, a highly relevant topic for the TMLR community as the field grapples with fair compensation for data creators and the sustainability of data markets.

S2. Solid theoretical contributions. The authors derive closed-form optimal mechanisms for five of six settings (M1, M2, M4, M5, M6) and provide bounds for M3. The construction of a potential function to handle path independence in the union setting (Lemma 2.3, Theorem 2.3) is technically elegant and may be of independent interest beyond this specific application.

S3. Clear economic insights. The paper provides interpretable results: Corollary 2.2 shows that no domain completely disappears due to diminishing returns and increasing marginal costs. The "lose-lose" theorem (Theorem 3.3) provides a clean characterization of inefficiencies introduced by data brokers. The surprising result that $Rev_4 = Rev_5$ offers interesting economic intuition about market power.

S4. Complete treatment of multiple market structures. Analyzing both revenue and welfare objectives across two-layer (with/without union) and three-layer settings provides a comprehensive picture. The comparison theorem (Theorem 2.4) nicely organizes the relationships.

S5. Computational tractability. Establishing polynomial-time algorithms (Proposition 2.1) makes the mechanisms practically implementable, which is often overlooked in theoretical mechanism design work.


**Weaknesses:**

W1. Strong modeling assumptions that may limit applicability. The representative-creator assumption (one creator per domain) is defended via aggregation (Section C.3), but the aggregation argument assumes homogeneous $\rho_k$ within domains. In practice, creators have heterogeneous production functions.
The assumption that $\mu_{ik}$ and $\gamma_{ik}$ are publicly known is strong—these transferability parameters are difficult to estimate in practice and may themselves be private information.
The independence assumption (Assumption 2.1) rules out correlated costs across domains, which seems unrealistic when creators face common market conditions.

W2. GenAI modeling choices are debatable. The assumption of "negligible" marginal cost for GenAI relative to human creation ignores substantial compute costs, especially for large models. As of 2024-2025, training and inference costs for frontier models are significant. The model assumes GenAI quality scales with human data via a simple power law, but recent work suggests more complex relationships (data quality, distribution shifts, etc.).

W3. Missing discussion of dynamic aspects. The model is static, but in reality, GenAI capabilities improve over time, potentially changing the entire market structure. How robust are these mechanisms to evolving $\gamma_{ik}$? Creator learning and strategic behavior over multiple interactions is not considered.

Lastly, mechanism M3 remains unsolved: The revenue-maximizing mechanism with a creator union lacks an explicit solution. Given that unions/collectives are increasingly relevant in practice (e.g., writers' guilds, content creator unions), this remains a gap.

**Requested Changes:**

It would be helpful if the authors could provide a discussion of the weaknesses outlined above.

---

> ### Author Response · Authors · 2026-01-07
>
> Thank you for the encouraging feedback and constructive questions. We believe our response below should help thoroughly address the reviewer's major questions, and are happy to engage with the reviewer if there is any further question.
>
> *Re Modeling assumptions*: This is a very insightful question. For ease of presentation, our main model adopted a few simplification assumptions as the reviewer pointed out. However, many of our results could be generalized beyond, as we elaborate below. First, we assume a common $\rho_k$ within each domain to reflect the economic intuition that creators operating in the same domain typically share similar production technologies. Importantly, our theoretical results (e.g., Theorem 2.1 and Proposition 2.1) do not rely on the homogeneity of $\rho_k$. In particular, if we allow the $i$-th creator in domain $k$ to have a heterogeneous parameter $\rho_k^i$, after our transformation the problem still reduces to an optimization problem that can be solved efficiently in polynomial time. We have added a discussion of heterogeneous $\rho_k^i$ in Appendix C.4.
>
> Regarding $\gamma$ and $\mu$, creators typically do not possess private information about cross-domain knowledge transfer, as they mainly focus on their own domain. But large content platforms, like YouTube, usually have sufficient resources to learn these parameters. Thus, treating these parameters as publicly known is a reasonable modeling abstraction in our setting.
>
> Finally, the independence assumption in Assumption 2.1 is not required for Mechanisms 3 and 4. These mechanisms remain valid even when costs are correlated across domains; see the discussion in Appendix C.5.2. More broadly, we believe our study of mechanism design without cost independence could be of independent interest.
>
> *Re GenAI modeling choices*: Training costs for GenAI models are largely fixed costs. A key feature of GenAI that is relevant for mechanism design in our setting is that, **relative to human creation**, its marginal cost is low. For example, for OpenAI’s flagship model GPT-5, generating one million tokens costs on the order of tens of dollars, which is negligible compared to the marginal cost for a human to generate similar length of writing.
>
> We appreciate  the reviewer's insightful comment about other possible scaling law forms. While the power law functional form may not be universally exact -- and it is perhaps unlikely to have such a universally correct law anyway -- we believe that, given the current literature, our adaption of the power-law relationship between performance and data is a well-motivated modeling assumption. First, though recent works start to investigate other forms of scaling, the power law form remains one of the most widely accepted forms for scaling laws of LLMs  [1]. Second, this form has also been adopted by recent works for modeling AI's creation capability [2]. Our goal is not to claim this functional form is universally exact, but rather to use a parsimonious model to highlight potential inefficiencies of GenAI in multi-layer markets. We hope this framework can motivate regulators to think more carefully about such inefficiencies and inspire future work that studies richer and more realistic settings.
>
> [1] Jared Kaplan, Sam McCandlish, Tom Henighan, Tom B Brown, Benjamin Chess, Rewon Child, Scott Gray,
> Alec Radford, Jeffrey Wu, and Dario Amodei. Scaling laws for neural language models. arXiv preprint
> arXiv:2001.08361, 2020.
>
> [2] Fan Yao, Chuanhao Li, Denis Nekipelov, Hongning Wang, and Haifeng Xu. Human vs. generative ai in
> content creation competition: Symbiosis or conflict? arXiv preprint arXiv:2402.15467, 2024.

---

> > ### Author Response · Authors · 2026-01-07
> >
> > *Re Dynamic aspects and Mechanism 3*: First, we appreciate this insightful comment and agree that  dynamic learning and creator adaptation over repeated interactions are important directions for future work. Indeed, we have added this as an open question in our Conclusion and Discussion section. Second, our  static model is a natural and necessary first step towards answering the above dynamic setting, and offers insights about what happens at the limit. Third, the following practical facts offers further justifications about the static setting. In reality, the emergence of substantially new GenAI models  typically occurs on the timescale of months or years. This allows the platform or regulator to first estimate $\gamma$ and then re-design or re-set up the mechanism accordingly.
> >
> > For unsolved Mechanism 3, this  is well-recognized difficulty in the literature: revenue-maximizing mechanism design in a multidimensional environment is an open problem [3,4,5], and, to date, no  closed-form characterization is known beyond a few very restricted special cases. Acknowleding this difficulty, we explicitly formulate the optimization problem for Mechanism 3 and provide both revenue and social-welfare upper and lower bounds in Theorem 2.4. Besides, the difficulty of Mechanism 3 stands in sharp contrast to Mechanism 4, which maximizes social welfare. Under welfare maximization, one can leverage a potential-function–based formulation to construct path-independent payments. This comparison highlights an important advantage of social-welfare maximization: it enables simple and practically implementable mechanisms. Meanwhile, our experiments show that the welfare-maximizing mechanism only incurs a minor revenue loss, further supporting its practical relevance. We add more discussion on $\mathcal{M}_3$ in Appendix C.6.3.
> >
> > [3] Patrick Briest, Shuchi Chawla, Robert Kleinberg, and S Matthew Weinberg. Pricing randomized allocations.
> > In Proceedings of the twenty-first annual ACM-SIAM symposium on Discrete Algorithms, pp. 585–597.
> > SIAM, 2010.
> >
> > [4] Constantinos Daskalakis. Multi-item auctions defying intuition? ACM SIGecom Exchanges, 14(1):41–75,
> > 2015.
> >
> > [5] Sergiu Hart and Noam Nisan. Approximate revenue maximization with multiple items. Journal of Economic
> > Theory, 172:313–347, 2017.

---

### Decision · Action_Editor_NWbK · 2026-02-02

**Recommendation:** Accept with minor revision

**Additional Comments:**

- In Section 2, the current setup appears to focus on a one-dimensional scenario regarding the content/cost parameters. It would be good to include a discussion of when the analysis breaks down if moving into a high-dimensional setting.

- In Section 3.3, regarding "the lose-lose effect in three-layer markets" and the implications from here. I wonder whether this conclusion depends on the model's specific setup; if so, adding a more detailed discussion would be useful. In particular, clarifying the limitations of the current modeling framework in the context of data curation.

- In Appendix C, some of the proofs regarding the IC and IR properties of the proposed mechanisms, e.g., Proof of Theorem 2.1 (top of page 24) are terse and could benefit from more explanations (this would also alleviate one reviewer's concerns regarding the required econ expertise to understand this paper).

**Audience:**

Yes

**Audience Explanation:**

This paper is situated at the intersection of generative AI and mechanism design. The topic and the insights drawn from the analysis are of interest to the TMLR audience.

**Claims And Evidence:**

Yes

**Claims Explanation:**

This paper examines the problem of data curation from an economic market-making perspective. This paper studies a mechanism design setting in which both human curators and GenAI curators co-exist, and derives optimal mechanisms under several different scenarios, including revenue-maximization, welfare-maximization, and when a third-party intermediary plays a role in the market-making process. In particular, GenAI curators are modeled as a transferable good, defined by a pair of transferability and capability parameters.

For each of these scenarios, the solution concept draws on solving a convex program to derive the optimal price-allocation mechanism. Additionally, several empirical simulations are conducted to provide interpretations of the solutions from this convex programming approach.

This paper is reviewed well by three independent referees, with most reviewers leaning towards acceptance. Therefore, I recommend acceptance with minor revision. I have several comments below that I hope will help further improve this paper.